

# Analysis of deformation bands associated with the Trachyte Mesa intrusion, Henry Mountains, Utah: implications for reservoir connectivity and fluid flow around sill intrusions

Penelope I.R. Wilson[1], Robert W. Wilson[2], David J. Sanderson[3], Ian Jarvis[1], Kenneth J.W. McCaffrey[4]

[1] Kingston University London, Penrhyn Road, Kingston-upon-Thames, KT1 2EE, UK
[2] BP Exploration Operating Company Limited, Chertsey Road, Sunbury on Thames, TW16 7LN, UK
[3] University of Southampton, University Rod, Southampton, SO17 1BJ, UK
[4] Durham University, Science Labs, Durham, DH1 3LE, UK

*Correspondence to*: Penelope I.R. Wilson (p.wilson@kingston.ac.uk)

**Abstract.** Shallow-level igneous intrusions are a common feature of many sedimentary basins, and there is increased recognition of the syn-emplacement deformation structures in the host rock that help to accommodate this magma addition. However, the sub-seismic structure and reservoir-scale implications of igneous intrusions remain poorly understood. The Trachyte Mesa intrusion is a small (~1.5 km$^2$), NE–SW trending satellite intrusion to the Oligocene-age Mount Hillers intrusive complex in the Henry Mountains, Utah. It is emplaced within the highly porous, aeolian Entrada Sandstone

Formation (Jurassic), producing a network of conjugate sets of NE–SW striking deformation bands trending parallel to the intrusion margins. The network was characterized by defining a series of nodes and branches, from which the topology, frequency, intensity, spacing, characteristic length, and dimensionless intensity of the deformation band traces and branches were determined. These quantitative geometric and topological measures were supplemented by petrological, porosity and microstructural analyses. Results show a marked increase in deformation band intensity and significant porosity reduction

with increasing proximity to the intrusion. The deformation bands are likely to impede fluid flow, forming barriers and baffles within the Entrada reservoir unit. A corresponding increase in Y- and X- nodes highlights the significant increase in deformation band connectivity, which in turn will significantly reduce the permeability of the sandstone. This study indicates that fluid flow in deformed host rocks around igneous bodies may vary significantly from that in the undeformed host rock. A better understanding of the variability of deformation structures, and their association with intrusion geometry, will have

important implications for industries where fluid flow within naturally fractured reservoirs adds value (e.g. hydrocarbon reservoir deliverability, hydrology, geothermal energy and carbon sequestration).

## 1 Introduction

Syn-emplacement deformation structures (faults, fractures and forced folds) provide the principal mechanism for the accommodation of magma at shallow crustal levels (e.g. Pollard and Johnson, 1973; Hansen and Cartwright, 2006; Senger et

al., 2015; Wilson et al., 2016). The thermal effects of intrusions on host rocks have been well studied (e.g. Jaeger, 1964;



Brooks Hanson, 1995; Rodriguez-Monreal et al., 2009; Senger et al., 2014; Aarnes et al., 2015; Gardiner et al., 2019), as too has been the hydrothermal fluid flow associated with their emplacement (e.g. Rossetti et al., 2007; Scott et al., 2015). However, the impact of these syn-emplacement deformation structures on post-emplacement fluid flow around intrusions is less well understood (Montanari et al., 2017). Our study is based on detailed kinematic, geometric and spatial analyses of

networks of deformation bands in a host-rock sandstone associated with emplacement of the Trachyte Mesa igneous intrusion, Henry Mountains, Utah (Fig. 1). The aims are to: (1) characterize deformation structures developed in the Entrada Sandstone around the intrusion; (2) track geometrical and topological changes towards the intrusion; and (3) discuss the implications of these changes for fluid flow in the host-rock sandstone.

Deformation bands are a common structure in many fluid and gas reservoir sandstones. In particular, weakly cemented and

highly porous sandstones are ideal candidates for the development of deformation bands (e.g. Aydin, 1978; Underhill and Woodcock, 1987; Shipton and Cowie, 2001; Fossen et al., 2007; Wibberley et al., 2007). Numerous studies have shown that deformation bands can have a significant influence on fluid flow (Antonellini and Aydin, 1994, 1995; Gibson, 1998; Fossen and Bale, 2007; Ballas et al., 2015). Due to their mode of formation (i.e. mainly cataclasis and compaction; Du Bernard et al., 2002; Fossen et al., 2007; Eichhubl et al., 2020) deformation bands tend to have lower permeabilities than their host

sandstones and, in turn, they negatively affect fluid flow (Sterlof et al., 2004; Fossen and Bale, 2007; Rotevatn et al., 2013). Porosity and permeability reductions due to deformation bands may significantly reduce reservoir connectivity by creating baffles to fluid flow (e.g. Taylor & Pollard, 2000; Sternlof et al., 2006; Torabi et al., 2008; Sun et al., 2011; Saillet & Wibberly, 2013) and even, in some cases, act as seals to hydrocarbon accumulations (e.g. Knipe et al., 1997; Ogilvie and Glover, 2001; Parnell et al., 2004).

Deformation bands can develop in most structural and tectonic settings, provided the host rock is susceptible to their formation (Fossen et al., 2007; Schultz et al., 2010; Soliva et al., 2013; Ballas et al., 2015). Deformation bands preferentially form in more poorly lithified layers within quartz arenite to arkosic sandstones (i.e. those lacking in lithics) at shallow depths (1–3 km; Fossen, 2010). This depth regime is coincident with the emplacement of many shallow-level igneous intrusions, and deformation bands have been reported to develop as accommodation structures associated with sills and laccoliths (e.g.

Morgan et al., 2008; Wilson et al., 2016, 2019; Westerman et al., 2017; Fig. 1). These deformation bands may have important implications on the compartmentalisation and fluid flow within reservoirs hosting intrusions. To date, few quantitative analyses have been carried out to analyse the deformation structures associated with movement and accommodation of mobile substrates such as salt (e.g. Antonellini and Aydin, 1995) or magmatic intrusions (e.g. Morgan et al., 2008; Senger et al., 2015).

A fracture network can be regarded as a system of fractures (including deformation bands) developed within the same rock volume and may be made up of multiple fracture sets (e.g. Fig. 2; Adler and Thovert, 1999; Peacock et al., 2016). Fractures are generally described by their geometry (e.g. orientation and length) and characteristic attributes (e.g. fracture type, morphology and mineral fill). These attributes may be used to define fracture sets (Priest, 1993; Adler and Thovert, 1999; Sanderson and Nixon, 2015, Procter and Sanderson 2018), which are often used to delineate distinct structural events within





the evolution of a wider fracture network. Wilson et al. (2016) used the term 'phases' to describe what are in effect the various fracture sets observed in the Trachyte Mesa study area. Through these attributes it was possible to gain a good understanding of the various fracture sets and networks, however the relationships between these systems (e.g. connectivity) requires further analysis. Sanderson and Nixon (2015, 2018) highlighted the use of 'topology' for describing the relationships between geometrical objects and, building on the work of Mauldon (1994), Manzocchi (2002), Rohrbaugh et al.

(2002) and Mäkel (2007), outlined a workflow for fracture analysis. This workflow has been applied as part of the present study.

## 2 Geological Setting

### 2.1 Trachyte Mesa Intrusion

The Trachyte Mesa intrusion is a small (1.5 km$^2$) satellite intrusion to the Mount Hillers intrusive complex in the Henry

Mountains, SE Utah (Fig. 1). The Henry Mountains intrusions are Oligocene in age (31.2 to 23.3 Ma; Nelson et al., 1992). These intrusions, therefore, post-date the minor Laramide deformation observed on the Colorado Plateau. The intrusions are emplaced within an approximately 3–6 km thick section of late Palaeozoic–Mesozoic strata overlying Precambrian basement; the Trachyte Mesa intrusion was likely emplaced at a palaeo-depth of ~3 km (Jackson and Pollard, 1988; Hintze and Kowallis, 2009).

Trachyte Mesa has an elongate, laccolithic geometry, trending NE–SW (Fig. 1b) with a thickness varying from 5–50 m (Morgan et al., 2008). The intrusion formed by the amalgamation and stacking of multiple thin (~1–5 m thick) sill sheets (Johnson and Pollard, 1973; Menand, 2008; Morgan et al., 2008; Wilson et al., 2016). It is generally concordant with the Entrada Sandstone Formation, within which it is emplaced (Johnson and Pollard, 1973; Morgan et al., 2008; Wetmore et al., 2009). The best exposures of the intrusion, contact and overlying host-rock can be found on the southern end of the north-

western lateral margin (Figs. 1 and 2).

### 2.2 The Entrada Sandstone Formation

The Entrada Sandstone Formation (part of the San Rafael Group) is Jurassic (Callovian) in age and is composed of a mixture of white cross-bedded sandstones, reddish-brown silty sandstones, siltstones, and shale beds (Aydin, 1978; Fig. 2a). The Entrada was deposited in an aeolian environment and extends over a vast area, making it the largest of the Colorado Plateau

ergs (Hintze and Kowallis, 2009). Entrada Sandstone is generally quartz-dominated (Aydin, 1978), although a subarkosic composition for rock units studied around the Trachyte Mesa intrusion may be a more appropriate lithological description. Calcite is the most common cement, although siliceous and pelitic cements are abundant in some layers (Aydin, 1978).

The Entrada Sandstone, being highly porous, is the ideal lithology for the formation of deformation bands (Fig. 2) and, as a result (along with the Lower Jurassic Navajo and Wingate sandstones, also found on the Colorado Plateau and

stratigraphically below the Entrada; Jackson and Pollard, 1988), has been the focus of several studies on such structures



(Aydin, 1978; Aydin and Johnson, 1978, 1983; Shipton and Cowie, 2001; Fossen and Bale, 2007; Fossen et al., 2007). These Jurassic sandstones form natural fluid carrier systems and reservoirs for hydrocarbon and $CO_2$ systems (Garden et al., 2001; Kampman et al., 2013). Although deformation bands are common throughout the Entrada Sandstone, local to the Trachyte Mesa intrusion, there appears to be a strong increase in deformation bands aligned sub-parallel to the intrusion margin. This

spatial and geometric correlation leads to the proposal that these structures formed directly in response to the emplacement of the intrusion (Wilson et al., 2016).

Deformation bands in the vicinity of Trachyte Mesa generally form as conjugate sets striking roughly NE–SW and individual bands are generally narrow (<0.5 mm; Figs. 1–3), though a few wider deformation zones have developed which are cored by principal slip surfaces. In contrast, much wider deformation band clusters (>20 cm) can be found hosted by the

Entrada Sandstone elsewhere on the Colorado Plateau (e.g. see figs. 1, 7 and 9 in Fossen and Bale, 2007).

## 3 Study area, sampling and analysis methods

### 3.1 Outcrop Traverse

Outcrop studies and rock samples were collected from a structural transect across the north-western lateral margin of the Trachyte Mesa intrusion (Figs. 2 and 3), in order to quantify the change in the network observed across the intrusion margin.

The study consists of six structural stations, relatively evenly spaced, from ~60 m outboard of the exposed intrusion margin (Station 1), and up over the monoclinal lateral margin, onto the top surface of the intrusion (Station 6) (Fig. 2). A suite of photographs was collected at each exposure (e.g. Fig. 2b–g). These photographs were subsequently used to map out the fracture networks at each station post-fieldwork (see appendices).

As roughly NE–SW trending deformation bands are the dominant structural orientation identified along the margin (Fig.1;

Wilson et al., 2016), care was taken to ensure that the surfaces photographed, and subsequently analysed, were oriented in a similar, optimal perpendicular (NW–SE) orientation in order to sample the network most appropriately. Due to this sampling technique, results will only be appropriate for analysing fluid flow across (perpendicular to) the intrusion margin, and further analysis may be necessary to understand flow parallel to the intrusion margin. It is acknowledged that by only carrying out studies in one orientation we are invoking an orientation bias into our results. However, choosing sections at a high angle to

the main orientation of the band intersections should minimise the bias in both the geometrical and topological parameters.

### 3.2 Sample Collection

A selection of rock samples was collected at each structural station (Fig. 3) in order to carry out hand-specimen and thin-section (i.e. petrological, porosity and microstructural) studies. Samples were oriented in the field in order to enable thin sectioning in a similar vertical, NW–SE oriented plane to the outcrop photograph/ scan surfaces. Ensuring that similarly

oriented sample areas are studied at all scales increases the chances of sampling the same fracture systems (i.e. NE–SW trending fracture networks) as observed in outcrop, and thus the resulting scalar statistics should be more appropriate.



### 3.3 Analysis Methods

Various analytical techniques have been proposed for the investigation of fracture networks (e.g. Walsh and Watterson, 1993; Berkowitz, 1995; Adler and Thovret, 1999). In this study the methods outlined by Sanderson and Nixon (2015) and

Procter and Sanderson (2018) have been applied. The method was described in detail by Sanderson and Nixon (2015), and only a brief summary is given here. The basic principle is outlined in Fig. 4, and comprises the mapping of a 2D fracture network, measuring fracture (or branch) lengths and quantities, and node counting (e.g. Fig. 5).

### 3.3.1 Fracture Network Map

Using the outcrop photographs, deformation band networks were mapped to the highest level of detail attainable from the

image resolution. Areas were then selected in order to sample the networks at each site. In areas of more heterogeneous deformation, multiple areas were sampled at different scales (ranging from 20–100 cm diameter circles) in order to capture the variability. Circular scan-lines/ areas were used rather than squares as these provide the least orientation bias, with an equal likelihood of sampling any given fracture orientation on a 2D surface (Mauldon, 1994; Rohrbaugh et al., 2002; Procter and Sanderson, 2018).

### 3.3.2 Measuring Lengths and Quantities

For each sample circle, the total number of fractures and total fracture length were recorded. In addition, the total number of fault branches (i.e. segments between intersecting fracture points or nodes) was also determined. Sanderson and Nixon (2015) argued that preference should be given to the use of branches as it is often difficult to recognise an individual, continuous fracture trace within a fracture network, whereas branches are uniquely identifiable. Furthermore, as exposures

and sampling areas are of finite size, many fractures extend beyond the sample area. Therefore, the frequency and length of fractures will be subject to sampling bias (Riley, 2005). By contrast, the length of branch lines is likely to be less censored, thus reducing this sampling bias issue.

Using sample area, total number of fractures (or branches) and total fracture (branch) length, a number of fracture network characteristics can be defined. These include: frequency (total number/ area); intensity (total length/ area); spacing (the

inverse of intensity, i.e. area/ total length); characteristic fracture length (mean length; total fracture length/ total number of fractures); and dimensionless intensity (multiplying fracture intensity by the characteristic length). Details on the derivation of these terms were provided by Sanderson and Nixon (2015).

### 3.3.3 Node Counting

A given fracture network consists of lines, nodes and branches (Figs. 4 and 5a–b). As outlined above, lines will consist of

one or more branches, with nodes (i.e. fracture intersections) at either end of each branch. Three main types of nodes exist: I-nodes (isolated fracture terminations within the host rock); Y-nodes (where one fracture terminates against another); and X-





nodes (where two fractures cross-cut one another). Within a sample area, a fourth type of node may also be recorded, where fractures intersect the outer perimeter of the sample area (termed E-nodes; Sanderson and Nixon, 2015). As discussed by Procter and Sanderson (2018), combining node counting with a measurement of intensity (usually $P_{21}$ – trace length per unit

area) provides a very efficient way to characterise both the geometry and topology of fracture networks.

The proportion of I-, Y- and X-nodes have been used by various authors to characterize a fracture network (e.g. Manzocchi, 2002, Mäkel, 2007) and the results plotted on a triangular diagram (Fig. 4). As the relative proportion of nodes will remain unchanged by any continuous transformations (i.e. scale changes and strains), this is termed a topological classification (Sanderson and Nixon, 2015).

**3.4 Thin section and porosity analysis**

Optical microscopy petrographical, porosity and microstructural analyses were carried out on thin sections cut from each hand specimen (e.g. Fig. 5c). Sections were impregnated with blue-dyed plastic resin in order to highlight porosity. Both compositional and porosity percentages were visually estimated using percentage estimation comparison charts (Bacelle and Bosellini, 1965; Tucker, 2001).

**4 Results**

**4.1 Fracture Analysis**

Six structural stations were analysed across an approximately 100-meter-long transect over the north-western intrusion margin (Fig. 2 and appendices). In order to maximise the area sampled at each scan station, data from multiple scan circles have been aggregated. Note, where scan circles overlap, data have been omitted from the totals to avoid duplication, e.g. at

Station 1 where all smaller scan circles lie within the larger circle. Results are summarised in Table 1 and Fig. 6. The values for Station 2, and to a lesser extent Station 1, are based on relatively few measurements. Procter and Sanderson (2018) recommended at least 30 nodes, so the data from Station 2 (7 nodes) should be considered unreliable, compared to the other stations where the number of nodes varies from 24 (Station 1) to 847 (Station 5).

Nodal populations for each station were recorded (Table 1, Fig. 6b) and plotted on triangular diagrams (Fig. 6c; after

Manzocchi, 2002). The outcrops studied show a clear dominance of I- and Y- nodes (Fig. 6), although the proportion of these nodes varies across the transect (Fig. 6b). Structural stations more distal to the intrusion (i.e. Stations 1 and 2) show approximately equal proportions of I- and Y-nodes (Fig. 6). The proportion of I-nodes decreases through Stations 3 to 4, where Y-nodes become dominant with proximity to the intrusion. At Stations 5 and 6 overlying the intrusion, I-nodes are negligible, and the nodal populations are dominated by Y- and X-nodes. These results reflect the overall increase in

connectivity of the conjugate deformation bands observed at the intrusion margin (Morgan et al., 2008; Wilson et al., 2016).

The abundance of deformation bands increases with proximity to the intrusion (Fig. 6d). The frequency of fractures ($P_{20}$, Fig. 6d) increases from ~10 m$^{-2}$ in Stations 1 and 2 to >>100 m$^{-2}$ above the intrusion (Stations 4–6). This is accompanied by an



increase in fracture intensity ($P_{21}$) from ~10 to 100 m$^{-1}$, despite a significant reduction in the length (<L>). This increase in intensity is accompanied by a decrease in length (<L> in Fig. 6d), and this is also seen in the branch data (<B> in Fig. 6e).

The net result is that there is little change in the dimensionless intensities of the traces ($4 < P_{22} < 8$) and branches ($0.8 < B_{22} < 2$), which can be interpreted in terms of the networks mainly becoming more intense towards the intrusion, but with their dimensionless geometry or "pattern" remaining fairly scale-invariant (but see comments on topology, in following paragraph).

The change in node types indicates a change in topology across the transect (Fig. 6b). Stations 1 and 2 plot in a region of the

I-Y-X triangle (Fig. 6c, which is where "tree-like" networks, with few cycles enclosing blocks typically develop (e.g. Fig. 2b, c), whereas the other stations are more dominated by connected nodes (Y and X), typical of networks with lots of cycles and deformation bands that enclose many small blocks (as seen in Figs. 3e,f and 5a,b). These topological changes can be monitored by several key parameters (Fig. 6f). The connections per line ($C_L$) increase to values of ~5 above the intrusion, and such values typify highly connected networks, although Sanderson and Nixon (2018) noted that $C_L$ for connected Y-

dominated networks is generally lower. They favoured the use of the number of connections per branch ($0 \leq C_B \leq 2$), which attains values close to 2 above the intrusion. The average degree <d> of the nodes (i.e. the number of branches that meet at a node) increases from <d> ≈ 2, typical of trees, to <d> ≈ 3, typical of many joint networks (e.g. Procter and Sanderson, 2018). Taken together, these topological parameters indicate greater connectivity of the deformation band networks as the intrusion is approached.

**4.2 Microstructural Analysis**

Thin section analysis of outcrop samples further shows a significant increase in deformation and fracture intensity with proximity to the lateral intrusion margin. Outboard (~40–60 m; Stations 1 and 2) from the lateral intrusion margin, samples show little sign of deformation and relatively high porosity (15–35 %; Fig. 7). Sample TMFS-1 – assumed to represent the host rock – is a well-sorted, medium- to fine-grained (~250 μm) sandstone, dominated by quartz (>80 %) and feldspar

(plagioclase and microcline; Fig. 8). In its seemingly undeformed state, using the classification of sandstones of Pettijohn et al. (1987), the host rock can be classified as a subarkose. Haematite can be seen coating quartz grains, whilst there is some weak sericitisation of feldspars due to alteration. No distinct cross-laminations are apparent when examining the thin section under the microscope, although weak layering is visible when viewing the whole thin section (Fig. 7a). The sample shows relatively high porosity, however zonal variations are apparent; an average porosity of ~25 %, rising to 30 % in places (Fig.

7b). The sample is relatively poorly cemented with patches of poikilotopic calcite spar (Fig. 8). No deformation bands have been identified in sample TMFS-1. However, the sandstone is relatively well compacted, with embayed contacts apparent at grain contacts.

TMFS-2 appears to sample a slightly coarser-grained bed within the sandstone horizon. Laminations are clearly visible at both the hand-specimen and thin-section scale (Figs. 3b and 7a). Thin-section analysis shows a well-sorted sample with





similar subarkose composition to TMFS-1. Sub-rounded grains suggest that this sandstone is relatively mature. Large patches of poikilotopic sparry calcite are present (Fig. 8). Similar to TMFS-1, this sample also shows no visible deformation bands. However, porosity is lower than that of TMFS-1, with an average porosity of ~20 % (Fig. 7). Calcite cementation reduces porosity significantly across the whole sample, with porosity for some laminations falling to <10 %.

Approaching the lateral intrusion margin (~20 m), sample TMFS-3 retains a high porosity of 30–35 % (Fig. 7), again
displaying only patchy poikilotopic calcite spar cementation (Fig. 8). The sample exhibits minor deformation, though only one deformation band was sampled. This band is not well developed and shows only weak deformation and grain crushing (cataclasis). The deformation band displays significant porosity reduction (<5 % porosity in the deformation band zone), with the majority of the porosity loss appearing to be due to the development a finer crystalline, equant calcite microspar cement (Fig. 8).

Moving onto the intrusion margin, sample TMFS-4, background (host rock) porosity is variable from lamina to lamina, although the average remains relatively high at 20–25 % (Fig. 7). TMFS-4 samples three discrete (~1 mm wide) deformation bands (Figs. 7 and 8). Microstructural analysis of these bands shows evidence for cataclasis (grain crushing and grain-size reduction). The grain size of the undeformed rock is medium to coarse (>250 μm), while within deformation bands this is significantly reduced (<50 μm). Although porosity is considerably reduced, microporosity (<5 %) is still apparent within
deformation bands. Calcite is also present within the deformation bands, accounting for some of the porosity reduction.

Immediately above the lateral intrusion margin, sample TMFS-5 (Fig. 7a) displays significant deformation zones. Background (host rock) porosity is lower than the less deformed samples described above (~15 %; Fig. 7b). This is due to greater compaction, as evidenced by the higher proportion of interlocking (more tightly packed) grains, embayed contacts and possible pressure solution with sutured grain contacts (Fig. 8). Calcite cementation within the background rock is patchy,
with calcite spar accounting for only 2–3 % of porosity reduction. Sampling several deformation bands, these appear more diffuse (up to 1 cm wide) than those sampled in TMFS-4. Although not well-established, distinct slip zones may be identified within deformation bands. Microporosity within the deformation bands is extremely low (~1 %). Clear cataclasis and associated grain-size reduction can be seen within the bands. Although larger grains are still present within the deformation bands, these show evidence for significant microfracturing and early development of sub-grain boundaries (Fig.
245 8).

Passing over the upper hinge zone of the monoclinal intrusion margin, sample TMFS-6 (Fig. 7a) shows a clear system of moderately-dipping cross-laminations. Again, a subarkose composition is apparent (quartz grains with lesser plagioclase- and microcline-feldspar). Embayed contacts are visible. Background porosity is significantly reduced in TMFS-6 compared to the other 5 samples, at 5 to 15 % (Fig. 7b). This is largely due to both compaction and greater calcite cementation (Fig. 8).
Multiple diffuse and discrete, anastomosing deformation bands are identifiable. Microporosity within the deformation bands is <2 %, again the result of cataclasis (grain-size reduction), compaction and cementation. Shearing of cross-laminations into deformation bands can be clearly seen (Figs. 7a and 8). Within deformation bands larger quartz and feldspar grains are still evident within a finer-grained cataclastic matrix. However, some of these larger grains have 'fuzzy' grain boundaries which





may reflect cataclasis along the boundary, while other grains show clear sub-grain boundaries parallel to deformation band
orientations (Fig. 8). Weakly developed slip planes are apparent within deformation bands, while at a grain-scale, shear can
also be identified (Fig. 8). Haematite is also incorporated into the matrix within deformation bands as a result of quartz grain
crushing. Note the brownish-staining of deformation bands in Figs. 7a and 8.

## 5 Discussion

### 5.1 Fracture Intensity and Topology Variations Across the Margin

A clear increase in fracture abundance can be observed across the intrusion margin. The quantitative analysis of fracture
attributes, such as intensity, frequency and dimensionless intensity, increase progressively across the intrusion margin (Fig.
6). In addition, analysis of nodal populations highlights the topological change across this same area. As fracture frequency
and intensity increase onto the intrusion, this is accompanied by an increase in Y (and to a lesser extent X) nodes. Manzocchi
(2002) and Sanderson and Nixon (2018) both linked changes in nodal populations to critical dimensionless intensity, and
percolation thresholds, using stochastic models.

Figure 6 shows the nodal distributions for this study overlain on contoured triplots defining lines of critical branch
dimensionless intensity ($B_{22}$). These contours represent the $B_{22}$ of a network with that topology when it is at the percolation
threshold (i.e. the limit or threshold at which the network is "unconnected"/ "connected"). If the network has a higher $B_{22}$
than that for its position in the contour plot then the network is considered connected; conversely if lower it is considered
unconnected. In Fig. 6 it is clear that the fracture networks at stations above the intrusion margin (i.e. Stations 4–6) are all
highly connected, with $B_{22}$ values well above the contour of critical branch dimensionless intensity within which the nodal
populations plot (cf. Sanderson and Nixon, 2018). In contrast, the fracture networks at scan stations outboard of the intrusion
(Stations 1 and 2) are clearly not connected; with $B_{22}$ values well below the contours of critical branch dimensionless
intensity. The $B_{22}$ value for TMFS-3, located ~20 metres away from the mapped intrusion margin, when considering the
nodal population triplot, suggest that the system is connected; however, the total value lies close to that of the percolation
threshold.

Nodal populations were also acquired at the hand-specimen scale (Fig. 3). Similar topological trends are apparent from these
hand-specimen samples; however, values appear more extreme at both end members (i.e. $B_{22} = \infty$ at Stations 1 and 2 where
samples contained no deformation bands, and $B_{22} > 2$ at Stations 4 and 5 overlying the intrusion) compared to the outcrop-
scale studies. These extreme end-member values may simply be due to sampling bias as part of the sample collection and,
had more samples been collected at each station, it is likely that total and average values would align better with those
obtained at outcrop.

Figure 9 shows schematic 3D block diagrams which compare the distribution of deformation band structures across the
Trachyte Mesa intrusion to forced folds above a normal fault (Ameen, 1990; Cosgrove, 2015). The variations in the
deformation band network geometry seen across the Trachyte Mesa intrusion margin (Wilson et al., 2016) are very similar to



those in the model for forced folds above a normal fault. The increase in fracture intensity, frequency and dimensionless intensity is also consistent with this model, with deformation increasing across the forced fold. Offsets are dominantly extensional, consistent with the forced-fold model.

The analogy to the growth of a normal fault is viable due to the mode of emplacement of the Trachyte Mesa intrusion through vertical stacking of sill sheets (Morgan, 2008; Wilson et al., 2016), which represent the uplifted footwall block. As the intrusion grows in size (by the incremental addition of sill sheets) this drives the shear localisation of deformation similar to that of a propagating normal fault (e.g. Ballas et al., 2015). However, as highlighted by Wilson et al. (2016), the order of stacking of sill sheets (over-, under-, mid-accretion; Menand, 2008) can significantly impact the style of syn-emplacement deformation within the overlying host-rocks (Fig. 9c). The transect in this study samples a section of the intrusion margin

which displays out-of-sequence (i.e. under- and mid-accretion; Menand, 2008) stacking, which leads to a broader monoclinal margin. In contrast, in a stepped margin (resulting from over-accretion of sill sheets), a more complex zonal variability in the fracture network and topology may be observed, rather than the gradual change seen for the monoclinal margin in this study.

### 5.2 Porosity and Microstructural Variations

Microstructural analysis of deformed samples shows dominantly brittle deformation with cataclastic flow and compaction

occurring within deformation bands. Despite significant porosity reduction from undeformed host rock (typically 20–30 %) to within deformation bands, micro-porosity of <2 % is still apparent within deformation bands. This porosity reduction is largely the result of cataclasis and compaction; however, calcite cementation also plays a significant role in many of the sample deformation bands. This order of magnitude change in porosity is consistent with many previous deformation band studies (e.g. Eichhubl et al., 2010; Sun et al., 2011; Ballas et al., 2015).

Deformation bands outboard of the intrusion margin (i.e. TMFS-1 to -3) appear to show dominantly compaction related deformation, with minor cataclasis as evidence for shear (Figs. 7 and 8). These would therefore be best categorised as pure and/or shear-enhanced compaction bands (PCBs and SECBs; Eichhubl et al., 2010; Ballas et al., 2015). As you move closer to the intrusion margin, more embayed contacts and evidence for pressure solution are observed (e.g. TMFS-4; Fig. 8). This may be an indication of increasing confining pressures and/or an increase in temperature related to proximity to the

intrusion. Deformation bands in samples collected from localities above the intrusion (i.e. TMFS-5 and -6) show significantly more evidence for cataclasis, crush-breccias and grain shearing (Fig. 8), highlighting the strain localisation in this domain (Fig. 9b). Although strain localisation is evident, few of the deformation bands analysed in this study exhibit well-defined principal slip surfaces or fault cores; these are however, more common in areas of the intrusion margin where sill sheets are stacked in a normal sequence and where strain is localised at individual sill-tip terminations (Fig. 9c; Wilson et

al., 2016).

As discussed by Ballas et al. (2015), these different deformation band types may each have distinctly different impacts on the overall permeability pathways of the sandstone. PCBs and SECBs may reduce the local permeability by two orders of magnitude, however the lack of cataclasis may negate these bands from forming barriers to flow (Rotevatn et al., 2009), but





may influence flow pathways (Sternlof et al., 2006). In contrast, cataclastic bands will also reduce the local permeability by
two, or more, orders of magnitude (Ballas et al., 2015), but may also significantly impede flow, forming barriers (e.g.
Ogilvie et al., 2001), and significantly impact flow pathways (Taylor and Pollard, 2000; Soliva et al., 2013). Therefore, in
addition to the topological variations outlined in Figure. 6, understanding the deformation band type is also an important
consideration.

In addition to porosity reduction due to deformation bands, a reduction in host-rock porosity is apparent within samples
TMFS-5 and TMFS-6, from sandstone beds overlying the intrusion (>25 % in samples TMFS-1 to -3 compared to <15 % in
samples TMFS-5 and -6). This reduction appears to be the result of greater compaction of grains and an increase in
cementation. The increased cementation observed in samples TMFS-5 and -6 could be related to hot fluids circulating
through the immediately surrounding host-rock strata during magma emplacement (e.g. the increase in Fe-staining seen in
TMFS-6; Fig. 7a). This is consistent with the observed porosity reduction and thinning of beds over the monoclinal intrusion
margin observed by Morgan et al. (2008).

### 5.3 Wider Implications

Shallow-level intrusions are a common feature of many hydrocarbon basins, including: the NE Atlantic margin (e.g. Malthe-
Sørrenssen et al., 2004; Hansen and Cartwright 2006); West of Shetland (e.g. Rateau et al., 2013; Gardiner et al., 2019); and
the southern and north-western margins of Australia (e.g. Holford et al., 2012; Mark et al., 2020). The present quantitative
study of fractures highlights the significant impact magma emplacement can have in highly porous siliciclastic reservoir
systems. Although only a small study, results show that fracture abundance and intensity increase markedly across the NW
margin of the Trachyte Mesa intrusion. The methods applied provide a means of quantifying this increase in deformation
intensity across an intrusive margin.

The deformation bands show significant porosity reduction that is most apparent in the sandstones overlying the intrusion.
The overall porosity reduction demonstrated in Fig. 7 would produce approximately an order of magnitude change in
permeability (e.g. assuming Kozeny-Carmen equation fundamentals; Civan, 2002, 2016), as observed in many reservoir
rocks. However, this assumes a homogeneous development of the grain-scale processes. Microstructural analysis suggests
that the porosity reduction is largely through localized development of deformation bands. These have been shown to start
away from the intrusion as poorly connected (or unconnected) networks, which might baffle and reduce fluid flow, but
probably to no great significance. In comparison, in the host rocks above the intrusion margin, the increase in Y- and X-
nodes highlights the significant increase in deformation band interconnectivity, which in turn will significantly reduce the
network connectivity and permeability pathways of the sandstone. Importantly, the formation of a connected network of such
bands may reduce permeability by several orders of magnitude (e.g Ballas et al., 2015).

The deformation aureole immediately bordering the intrusions has not been analysed as part of this study. However, this is
an important factor to consider when assessing the likely impact that intrusion-related deformation may have on a wider
reservoir system. At Trachyte Mesa, deformation bands decrease markedly from ~5 to 10 m above the intrusion margin



(although limited outcrop extent prevents a more detailed quantification of this). However, considering an intrusion the size of Trachyte Mesa (~1.5 km$^2$), this ~10 m thick zone of deformation may reduce significantly the exploitable reservoir volume. In addition to reducing the overall connected network of the reservoir, as the deformation bands also show strong

alignment to the intrusion margin (Wilson et al., 2016), an anisotropy to any permeability pathways around the intrusion will also be likely (similar to that in a fault zone, e.g. Farrell et al., 2014) to further impact connectivity beyond just these high intensity zones.

Although additional analyses are required in order to understand the 3D connectivity of these fracture systems, the present 2D analytical study goes a long way to establishing the connectivity of deformation bands in the host rocks to the Trachyte

Mesa intrusion. The more pertinent issue is understanding the effects this connectivity could have on permeability within the host rocks. This study emphasises the potential importance of understanding the impact of syn-emplacement deformation to localised fluid flow around igneous intrusions. Gaining a better understanding of these emplacement-related deformation structures will therefore have important implications for fluid flow, hydrocarbon reservoir connectivity/ deliverability, hydrology, geothermal energy and CO$_2$ sequestration (e.g. Garden et al., 2001; Holford et al., 2012; Tueckmantel et al.,

2012; Scott et al., 2015; Weis, 2015) in reservoirs and basins hosting igneous intrusions. Additionally, quantitative field studies, such as the one carried out here, are essential to improve and constrain laboratory and numerical models of intrusion emplacement mechanisms and associated deformation (e.g. Kavanagh et al., 2006; Montanari et al., 2017; Bertelsen et al., 2018; Galland et al., 2020).

## 6 Conclusions

Deformation structures vary in style and intensity across the lateral "monoclinal" margin of the Trachyte Mesa intrusion, but there is a clear relationship between deformation and proximity to the intrusion margin. This has led a number of authors to propose that these deformation structures developed in response to emplacement of the intrusion (e.g. Johnson and Pollard, 1973; Morgan et al., 2008; Wilson et al., 2016; this study).

Although only a small study, our results show that deformation bands increase in abundance and intensity across the NW

margin of the Trachyte Mesa intrusion. The methods applied provide a means of quantifying this increase in deformation intensity across the intrusive margin. Furthermore, the application of topologic analysis (in the form of nodal analysis) provides a means of understanding the network connectivity of deformation structures, and thus their negative impact on reservoir permeability. The increase in Y- and X-nodes with proximity to the intrusion likely creates a baffle or barrier to flow perpendicular to the intrusion margin, as well as potentially forming non-producible/ penetrable reservoir zones.

This study highlights that fluid flow in deformed host rocks around igneous bodies may vary significantly from that of the undeformed host-rock reservoir. Therefore, a better understanding of the variability of deformation structures, and their association with intrusion geometry, will have important implications for industries where fluid flow within naturally



fractured reservoirs adds value (e.g. hydrocarbon reservoir deliverability, hydrology, geothermal energy and carbon sequestration).

**Acknowledgements**

The authors would like to thank Casey Nixon for advice during the development of the work. P. Wilson acknowledges Kingston University London for PhD funding and laboratory access that supported this research. D. Sanderson acknowledges support from a Leverhulme Emeritus fellowship during the development of this work.

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



**Figure 1: Geological setting and study area. (a)** Simplified maps showing location of the Henry Mountains intrusive complex and the Trachyte Mesa intrusion. Intrusion outlines adapted from Morgan et al. (2008). **(b)** Contoured (20 m intervals) aerial image of the Trachyte Mesa area showing the intrusion outline (yellow) and study area. Dashed lines in the SW depict the sub-surface extent of the intrusion, as defined by Wetmore et al. (2009) from magnetic resistivity data. Blue dots show field localities visited as part of wider reconnaissance studies (Wilson et al., 2016). **(c)** Satellite photograph (© Google Earth) of the study area (NW margin of the Trachyte Mesa intrusion). Structural stations for fracture studies indicated by numbered red dots. Yellow dashed lines show outcrop exposure of sill sheet terminations. **(d)** Field photograph showing monoclinal geometry of the NW intrusion margin. Note blocky, red Entrada Sandstone units concordant with the underlying intrusion top surface, and stacked intrusive sheets below (sheet terminations highlighter in yellow). Structural Stations 2 – 6 are indicated (red dots). Viewpoint location shown in (c). Contoured equal-area stereoplot shows poles to planes for deformation bands measured across the NW-margin of the Trachyte Mesa intrusion (from Wilson et al., 2016).




Figure 2: Sampling traverse across lateral margin of the Trachyte Mesa intrusion. (a) Panoramic photograph of study area. Note red, cross-bedded Entrada Sandstone unit, and Trachyte Mesa intrusion outcropping to the right. All structural stations lie within the same more massive, cross-bedded unit. (b) – (g) Overview outcrop photographs for each station.







**Figure 3: Fracture analysis of hand specimens. (a) – (f) Bedding-normal cut surfaces with hand-specimen photographs showing fracture analysis circles for each structural station (note, sample numbers correspond to their respective station). Fracture analysis was carried out on freshly cut surfaces. Circular scans show the fracture network and associated I-, Y-, X- and E-nodes (for explanation of terminology see Fig. 4 and Sanderson and Nixon, 2015). Statistics show total number of branches (N), total fracture length (SL), fracture density/ intensity ($P_{21}$), and branch dimensionless intensity ($B_{22}$).**






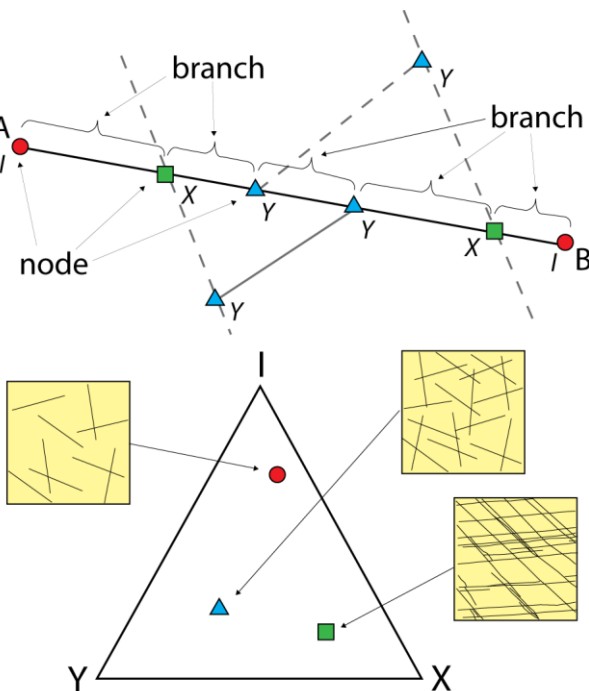

**Figure 4: Schematic image outlining the principal method applied for fracture analysis (Sanderson and Nixon, 2015). Branches**
**and nodes are shown on fracture trace (A–B): I-nodes (red circles); Y-nodes (blue triangles); X-nodes (green squares). Proportions**
**of I-, Y- and X-nodes may be plotted on a ternary plot to visualise different fracture network types (after Manzocchi, 2002).**





**Figure 5: Example of fracture analyses undertaken at different scales in this study, sample TMFS-5. (a) Outcrop photograph with superimposed fracture analysis circle showing the fracture network and associated I-, Y-, X- and E-nodes. (b) Hand specimen analysis (see Fig. 3). (c) Whole thin-section photograph (taken using flatbed scanner) and plain-polarised light (PPL) photomicrograph. Optical microscopy petrographical, porosity and microstructural analyses were carried out on thin sections cut from each hand specimen. Sections were impregnated with blue resin to highlight porosity.**





**Figure 6: Nodal and fracture analysis results. (a) Schematic diagram showing relative location of each structural station across the monoclinal intrusion margin. (b) Bar chart showing spatial variation in nodal populations. (c) Triangular plot showing ratio of I-, Y- and X-nodes for total values for each station. Contours represent the branch dimensionless intensity ($B_{22}$) of a network with that nodal topology when it is at the percolation threshold (i.e. the limit or threshold at which the network is "unconnected"/**

**"connected"). If the network has a higher $B_{22}$ than in the contour plot then the network is considered connected; conversely if lower it is considered unconnected. As the $B_{22}$ values for Stations 1 and 2 (see Table 1) are below the contour values, these deformation band networks are clearly unconnected, whereas Stations 4, 5 and 6 are well above the contour values and thus may be considered well connected. For more details on the triplot template, see Sanderson and Nixon (2018). (d)-(f) Summary log-linear plots showing various fracture attributes at each station (see Table 1 for values). <L>: Mean line length (m; total line length/**

**number of lines); $P_{20}$: line frequency; $P_{21}$ ($m^{-2}$; number of lines/ sample area): line intensity ($m^{-1}$; total line length/ sample area); $P_{22}$: line dimensionless intensity (multiplying line intensity by the mean length); <B>: Mean branch length (m; total branch length/ number of branches); $B_{20}$: branch frequency ($m^{-2}$; number of branches/ sample area); $B_{21}$: branch intensity ($m^{-1}$; total branch length/ area); $B_{22}$: branch dimensionless intensity (multiplying branch intensity by the mean length); <d>: average degree of nodes (the number of branches that meet at a node); $C_L$: connections per line; $C_B$: connections per branch.**








(a)

(b)

Figure 7: Porosity variability across the intrusion margin. (a) Whole thin-section photographs (flatbed scans) for each structural station. Blue dye denotes porosity in each sample. (b) Plot showing variability in porosity observed in this study for each station. Porosity percentages were estimated using visual comparison charts (Bacelle and Bosellini, 1965, Tucker, 2001). Note, a similar porosity reduction trend has been observed previously (see fig. 8 in Morgan et al., 2008). DB: Deformation Band; X-lam: Cross-lamination.

**Figure 8: Thin section photomicrographs for each sample. Note overall decrease in porosity and increase in cataclasis (within deformation bands) and calcite cementation from sample TMFS-1 through to TMFS-6. Cal: Calcite spar; DB: Deformation Band; Fe: Iron staining; Pl: Plagioclase Feldspar; Qtz: Quartz. Red arrows highlight zones of pressure solution, embayed contacts, and grain shear.**





**Figure 9: Schematic 3D block diagrams and cross sections comparing the distribution of deformation structures. (a) A forced fold above a normal fault (modified after Ameen, 1990 and Cosgrove, 2015). (b) Deformation bands across the Trachyte Mesa intrusion (this study). (c) Cartoon showing varying deformation styles and distribution in relation to the order of sill sheet stacking (Wilson**






et al., 2016). In (a) the fold is divided into zones (see inset table) depending on the level of strain normal ($e_z$) and parallel to the layer ($e_y$ parallel to the fold hinge and $e_x$ normal to it). Note: extension is negative and contraction positive. Coloured zones highlighted in (b) are solely for visual purposes and do not correspond to the strain zones defined in (a).

| | | | TMFS_1 | TMFS_2 | TMFS_3 | TMFS_4 | TMFS_5 | TMFS_6 | *units* |
|---|---|---|---|---|---|---|---|---|---|
| | | | **1** | **2** | **3** | **4** | **5** | **6** | |
| **Nodes** | I-nodes | **I** | 11 | 4 | 15 | 48 | 102 | 2 | |
| | Y-nodes | **Y** | 9 | 3 | 44 | 83 | 528 | 435 | |
| | X-nodes | **X** | 4 | 0 | 22 | 19 | 217 | 114 | |
| | # Nodes | **\|N\|** | 24 | 7 | 81 | 150 | 847 | 551 | |
| **Topology** | # Branches | **\|B\|** | 27 | 6.5 | 117.5 | 186.5 | 1277 | 881.5 | |
| | # Lines | **\|L\|** | 10 | 3.5 | 29.5 | 65.5 | 315 | 218.5 | |
| | Average degree | **<d>** | 2.25 | 1.86 | 2.90 | 2.49 | 3.02 | 3.20 | |
| | Proportion (Y+X) | **P(x+y)** | 0.70 | 0.43 | 0.82 | 0.70 | 0.91 | 1.00 | |
| **Line** | Frequency | **P20** | 23.55 | 3.59 | 57.00 | 157.0 | 461.5 | 843.0 | *m-2* |
| | Intensity | **P21** | 9.79 | 3.77 | 13.40 | 26.72 | 49.07 | 80.87 | *m-1* |
| | Dimensionless intensity | **P22** | 4.07 | 3.96 | 3.15 | 4.55 | 5.22 | 7.76 | |
| | Ave. Line length | **<L>** | 0.42 | 1.05 | 0.24 | 0.17 | 0.11 | 0.10 | *m* |
| | Connections per line | **C_L** | 3.70 | 1.71 | 4.44 | 3.19 | 4.93 | 5.03 | |
| **Branch** | Frequency | **B20** | 82.0 | 6.7 | 225.0 | 453.3 | 1932.8 | 3401.1 | *m-2* |
| | Intensity | **B21** | 9.79 | 3.77 | 13.40 | 26.72 | 49.07 | 80.87 | *m-1* |
| | Dimensionless intensity | **B22** | 1.17 | 2.13 | 0.80 | 1.57 | 1.25 | 1.92 | |
| | Ave. Branch length | **<B>** | 0.12 | 0.56 | 0.06 | 0.06 | 0.03 | 0.02 | *m* |
| | Connections per branch | **C_B** | 1.78 | 1.38 | 1.87 | 1.76 | 1.94 | 2.00 | |
| | Proportion of I-nodes | **P(I)** | 0.46 | 0.57 | 0.19 | 0.32 | 0.12 | 0.00 | |
| | Proportion of Y-nodes | **P(Y)** | 0.38 | 0.43 | 0.54 | 0.55 | 0.62 | 0.79 | |
| | Proportion of X-nodes | **P(X)** | 0.17 | 0.00 | 0.27 | 0.13 | 0.26 | 0.21 | |


**Table 1: Summary table showing total values for each structural station. Total values do not include hand specimens due to their small size making estimates of frequency and intensity unreliable, though trends for samples do match those for outcrops in this study. For details on individual scan circles see appendices.**



**Appendix**

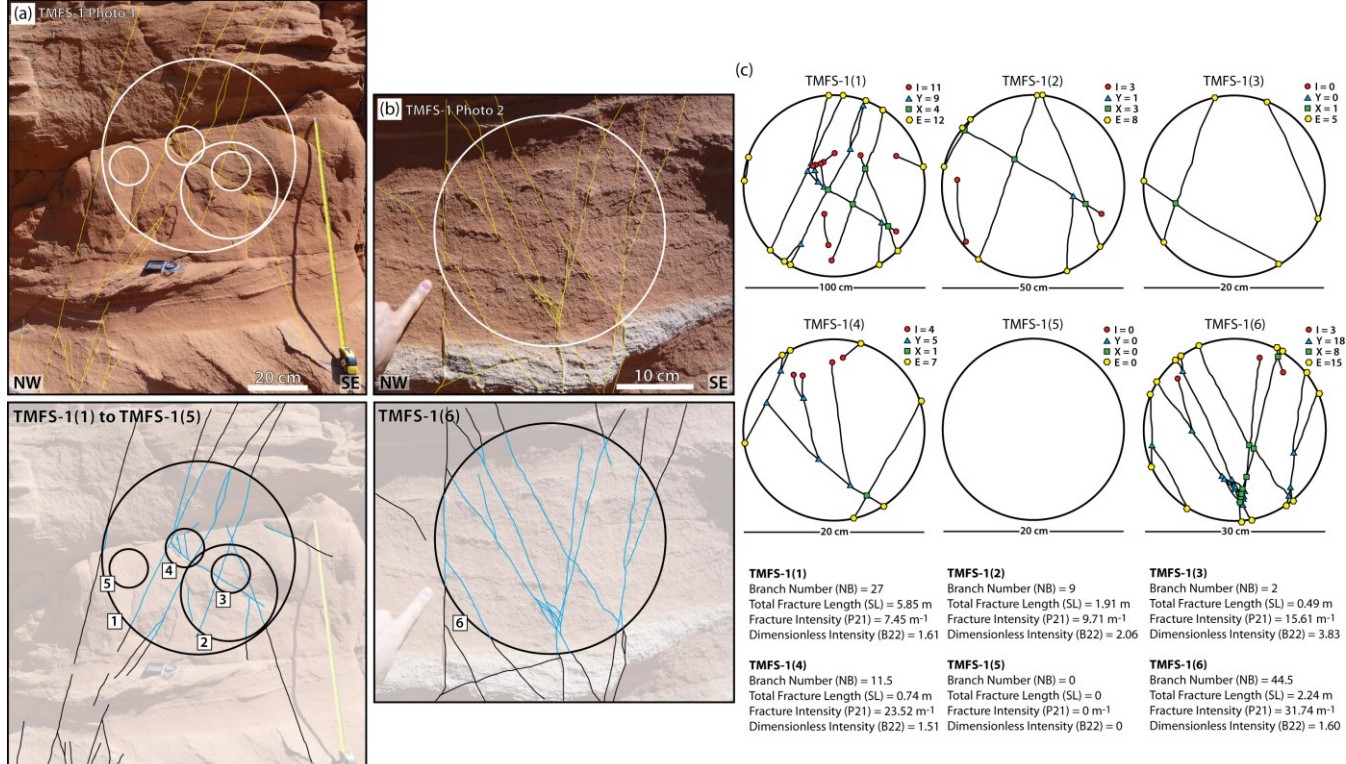



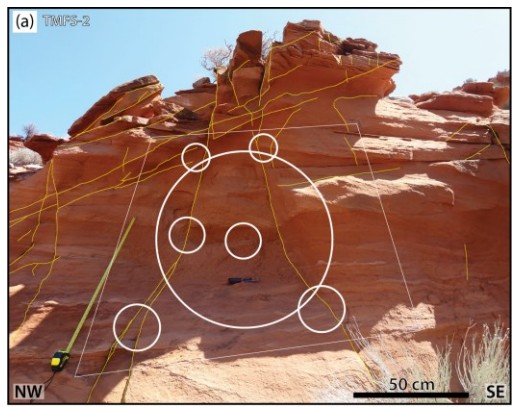

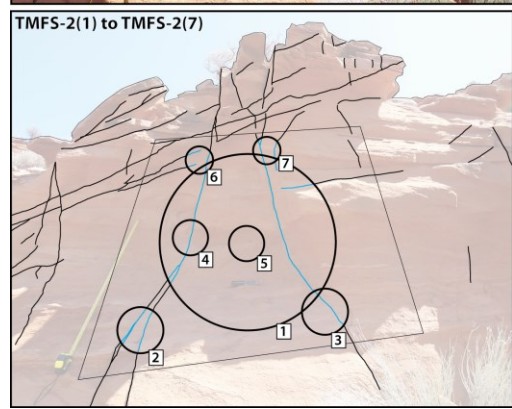

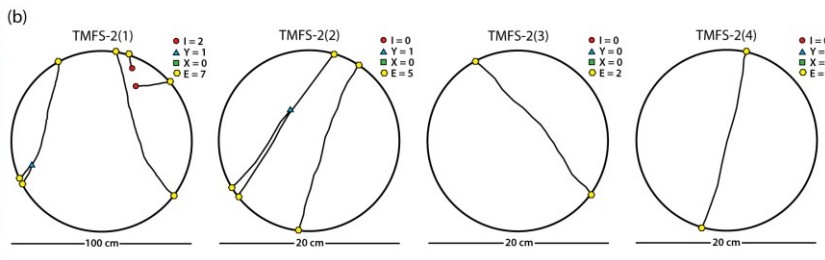

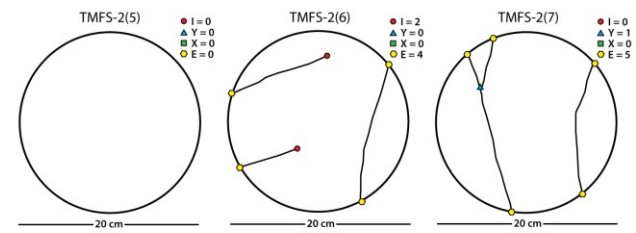

**TMFS-2(1)**
Branch Number (NB) = 2.5
Total Fracture Length (SL) = 1.96 m
Fracture Intensity (P21) = 2.50 m⁻¹
Dimensionless Intensity (B22) = 1.96

**TMFS-2(2)**
Branch Number (NB) = 1.5
Total Fracture Length (SL) = 0.59 m
Fracture Intensity (P21) = 18.65 m⁻¹
Dimensionless Intensity (B22) = 7.28

**TMFS-2(3)**
Branch Number (NB) = 0
Total Fracture Length (SL) = 0.20 m
Fracture Intensity (P21) = 6.32 m⁻¹
Dimensionless Intensity (B22) = 0

**TMFS-2(4)**
Branch Number (NB) = 0
Total Fracture Length (SL) = 0.20
Fracture Intensity (P21) = 6.40 m⁻¹
Dimensionless Intensity (B22) = 0

**TMFS-2(5)**
Branch Number (NB) = 0
Total Fracture Length (SL) = 0 m
Fracture Intensity (P21) = 0 m⁻¹
Dimensionless Intensity (B22) = 0

**TMFS-2(6)**
Branch Number (NB) = 1
Total Fracture Length (SL) = 0.34 m
Fracture Intensity (P21) = 10.72 m⁻¹
Dimensionless Intensity (B22) = 3.61

**TMFS-2(7)**
Branch Number (NB) = 1.5
Total Fracture Length (SL) = 0.39 m
Fracture Intensity (P21) = 12.38 m⁻¹
Dimensionless Intensity (B22) = 3.21






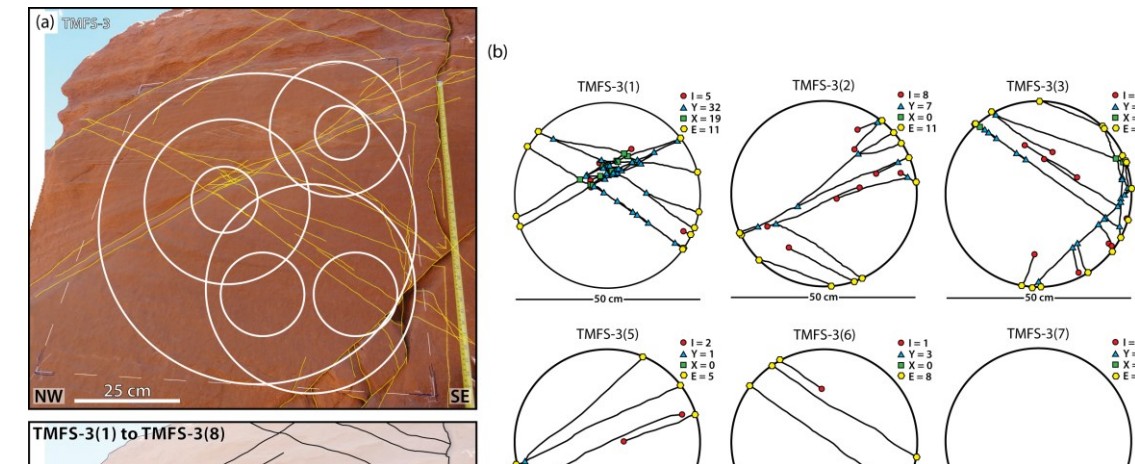

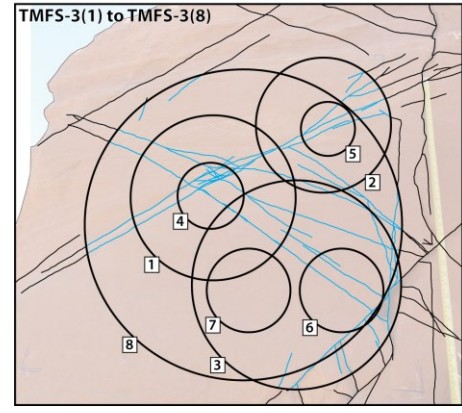

**TMFS-3(1)**
Branch Number (NB) = 88.5
Total Fracture Length (SL) = 3.46 m
Fracture Intensity (P21) = 17.63 m⁻¹
Dimensionless Intensity (B22) = 0.69

**TMFS-3(2)**
Branch Number (NB) = 14.5
Total Fracture Length (SL) = 2.47 m
Fracture Intensity (P21) = 12.55 m⁻¹
Dimensionless Intensity (B22) = 2.13

**TMFS-3(3)**
Branch Number (NB) = 42.5
Total Fracture Length (SL) = 2.94 m
Fracture Intensity (P21) = 14.99 m⁻¹
Dimensionless Intensity (B22) = 1.04

**TMFS-3(4)**
Branch Number (NB) = 67
Total Fracture Length (SL) = 1.38 m
Fracture Intensity (P21) = 43.82 m⁻¹
Dimensionless Intensity (B22) = 0.90

**TMFS-3(5)**
Branch Number (NB) = 2.5
Total Fracture Length (SL) = 0.64 m
Fracture Intensity (P21) = 20.35 m⁻¹
Dimensionless Intensity (B22) = 5.20

**TMFS-3(6)**
Branch Number (NB) = 5
Total Fracture Length (SL) = 0.55 m
Fracture Intensity (P21) = 17.57 m⁻¹
Dimensionless Intensity (B22) = 1.94

**TMFS-3(7)**
Branch Number (NB) = 0
Total Fracture Length (SL) = 0 m
Fracture Intensity (P21) = 0 m⁻¹
Dimensionless Intensity (B22) = 0

**TMFS-3(8)**
Branch Number (NB) = 117.5
Total Fracture Length (SL) = 8.67 m
Fracture Intensity (P21) = 11.03 m⁻¹
Dimensionless Intensity (B22) = 0.81




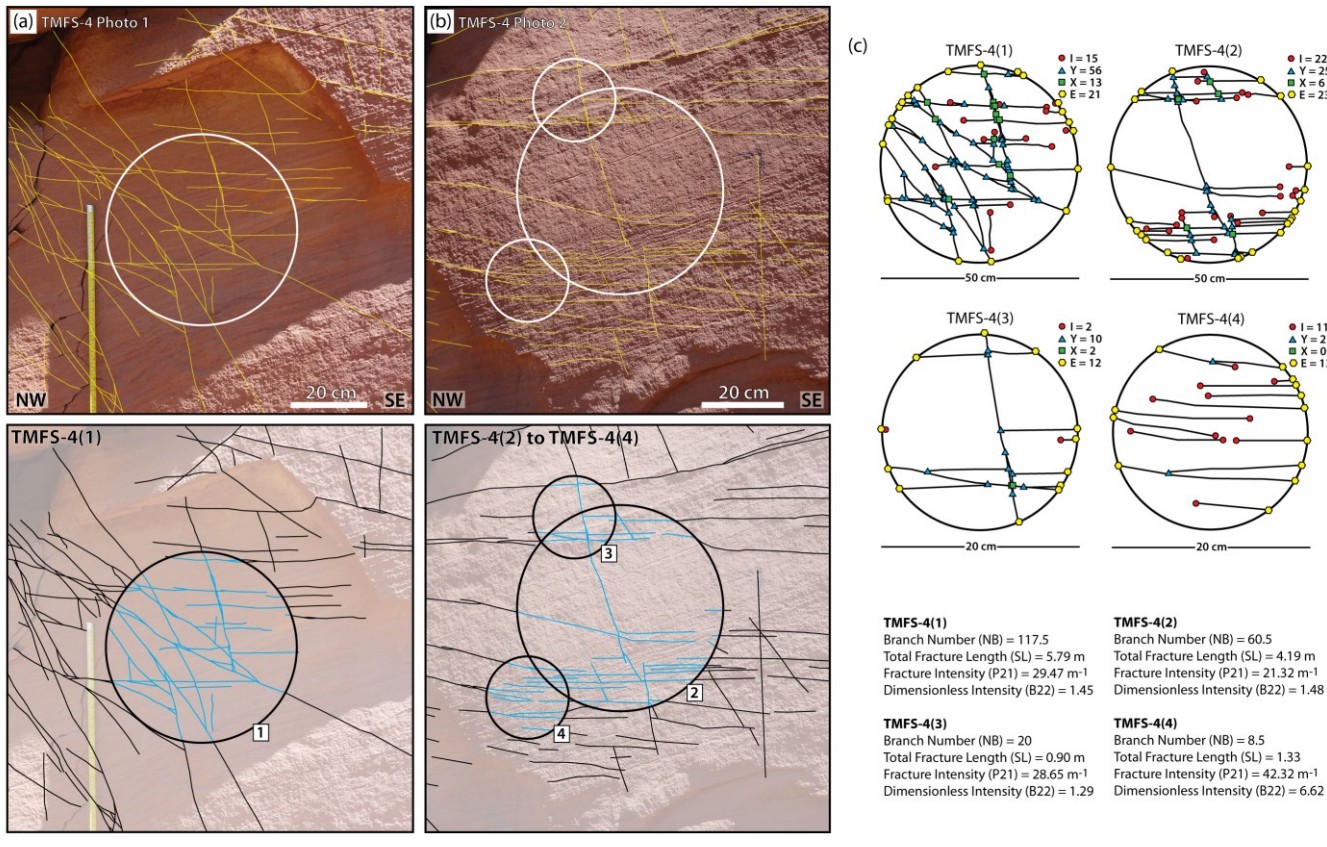

**TMFS-4(1)**
Branch Number (NB) = 117.5
Total Fracture Length (SL) = 5.79 m
Fracture Intensity (P21) = 29.47 m⁻¹
Dimensionless Intensity (B22) = 1.45

**TMFS-4(2)**
Branch Number (NB) = 60.5
Total Fracture Length (SL) = 4.19 m
Fracture Intensity (P21) = 21.32 m⁻¹
Dimensionless Intensity (B22) = 1.48

**TMFS-4(3)**
Branch Number (NB) = 20
Total Fracture Length (SL) = 0.90 m
Fracture Intensity (P21) = 28.65 m⁻¹
Dimensionless Intensity (B22) = 1.29

**TMFS-4(4)**
Branch Number (NB) = 8.5
Total Fracture Length (SL) = 1.33
Fracture Intensity (P21) = 42.32 m⁻¹
Dimensionless Intensity (B22) = 6.62



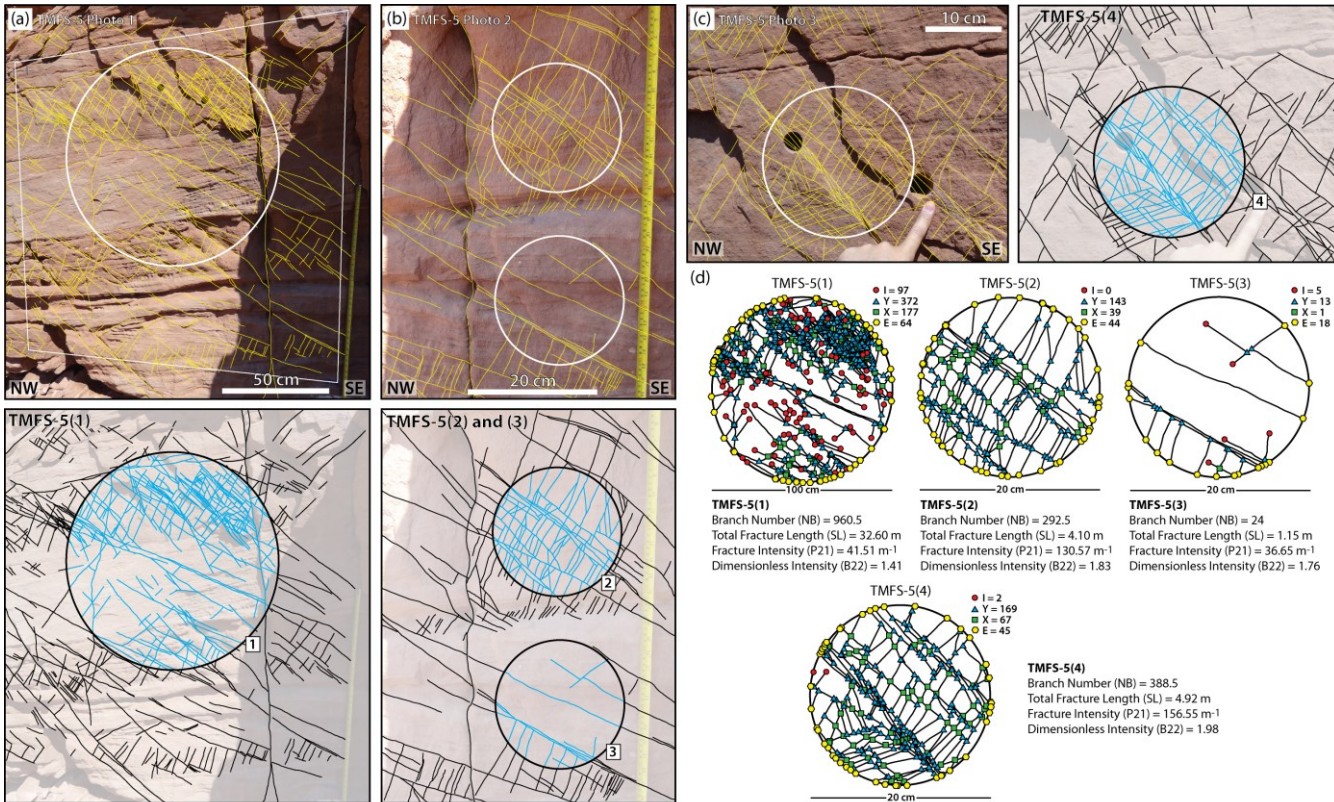



