# Peer review of "Analysis of deformation bands associated with the Trachyte Mesa intrusion, Henry Mountains, Utah: implications for reservoir connectivity and fluid flow around sill intrusions"

_Solid Earth, 2020_

## Short Comment (SC1) · 2 Jul 2020

Dear authors,

This, in my opinion, is an excellent paper and I thoroughly enjoyed reading through it. How fluid flow around intrusions is affected by syn-emplacement structures is something I've been pondering for a while so I'm glad to see you've already taken a good step towards understanding this. The science is solid and the paper well-written and beautifully illustrated. I just wanted to put forth some ideas I had that may be worth

exploring and could perhaps, in a small way, strengthen the paper even further:

1) During such monoclinal folding above intrusion tips, there is plenty of evidence to suggest that outer-arc stretching promotes tensile failure and that these fractures are exploited by intruding magma and facilitate the transgression of inclined limbs. As far as I'm aware, no such inclined sill limbs are seen at Trachyte Mesa. Are such tensile fractures observed though near the sill margin where flexure is greatest? If not, why might there be no outer-arc stretching structures?

2) I suspect this comment may partly answer the questions in my previous comment but...it seems that magma emplacement was accommodated by both uplift and 'compaction' of the Entrada Sandstone, perhaps with the latter significant enough that sufficient outer-arc stresses could not build-up? I think it worth highlighting that the synchronous occurrence of these space-making mechanisms means that the amplitude/volume of the is less than the thickness/volume of the intrusion; i.e. deformation could not be described as purely elastic. This is yet further evidence that inverting ground deformation data, to recover intrusion geometries and locations, using elastic models is likely inappropriate in some scenarios.

3) Trachyte Mesa comprises multiple sheets. A key outstanding question I think in these sorts of intrusions is how long did each injection event last and what was the duration between each intrusion? This is very hard to get at with geochronology if time differences are small but, and this may be my ignorance speaking here, is there anyway to use deformation band / fracture sets to identify different pulses and reconstruct the strain rate from their microstructures (assuming strain rate effectively equals intrusion rate)?

I look forward to seeing the published version!

Kind regards, Craig Magee

---

## Referee Comment (RC1) · Anonymous Referee #1 · 3 Aug 2020

Dear Authors,

This pre-print provides a nice example of using quantitative fracture analysis at a variety of scales to assess the impact that the growth of a shallow igneous intrusion (made up of stacked sills) has on the abundance and topology of a deformation band network in a porous sandstone. The study should be of interest to geologists assessing fluid flow (hydrocarbon, water, carbon dioxide) in such environments worldwide as well as those interested in wall-rock processes associated with accommodating shallow intrusion volumes. The manuscript is well written and thoroughly illustrated.

[Figure]

Specific Comments:

One specific area that I would like to see developed a bit further is the conceptual model (Fig 9) presented in the final discussion (Sections 5.1 and 5.2). You compare the patterns of deformation you observe at Trachyte Mesa (9b) to those in a forced fold above a normal fault (9a). I hope you can expand this discussion to address how differences between the processes might be reflected in the patterns of deformation observed. The sills have intruded laterally underneath the entire mesa (i.e. see Fig. 12 in Morgan et al., 2008), whereas the forced fold formed over either an upward or laterally propagating fault tip (e.g. White and Crider, 2006). One might therefore expect a structural density greater than the background above the intrusion but not above the footwall of the normal fault. Although your sample transect doesn't appear to extend far enough to directly address this question, it would be nice if you speculated on it a bit in the discussion. Perhaps there might also be differences in the orientations of structures?

I also found it a bit confusing that you mention that Sanderson and Nixon (2015) argue for use of branch attributes (vs length) when characterizing fracture networks (lines 142-143), but then proceeded to focus on length attributes in your discussion of the results (lines 186-189). Perhaps you should drop this point from the earlier discussion or else recast your results to emphasize branch attributes?

Other Minor/Technical Corrections:

Line 45 an 'n' is missing from Sternlof;

Line 56 'implication for' rather than 'on';

Lines 114-116. Please add some estimate of the average (central tendency) orientation of deformation bands and analysis sections here.

Fig 1 caption. Change 'outlines' to 'areas'.

Fig. 6c. It would be nice to have the B22 values for each point annotated somewhere

on the plot, either in the legend or with the point labels.

Figure 6 caption: Instead of using "log-linear", which implies that the station numbers have quantitative meaning, reword to "Summary plots showing the log of various fracture attributes at each station"

[Figure]

---

## Author Comment (AC1) · 7 Aug 2020

Thank you Craig for your positive commentary on the paper, and for yours ideas, which we will certainly take into account in our revisions.

Below are responses to the 3 points raised.

(1) The monoclinal fold model that you describe does not appear to be a good representation of the deformation along the intrusion margin at Trachyte Mesa, because most of the deformation structures observed are shear rather than tensile fractures.

[Figure]

The strain is, therefore, not taken up by outer-arc extension, but by conjugate shear faults/deformation bands, e.g. compare images C and D in Figure 21.18 in Ramsay and Huber (Modern Structural Geology, Vol. 2 – Folds and Fractures).

Inclined sill sheets are observed at the Trachyte Mesa intrusion, but only along intrusion margins where over-accretion of sill sheets is observed, resulting in a stepped-geometry. Along such margins, due to a two-stage emplacement mechanism for individual sheets – with initial propagation of a thin sill sheet, followed by secondary inflation – syn-emplacement faults develop at the inflating sill tip during inflation (see Wilson et al., 2016). As highlighted in Wilson et al. (2016), if these faults dip towards the intrusion (i.e. reverse movement), magma propagation can occur upwards along the fault plane. However, magma is unable to propagate along 'normal' faults, as the stress is non-optimal for magma propagation as the fault/ fractures remain closed. See Fig. 15 in Wilson et al. (2016).

We do not observe any inclined sill sheets along the transect studied in this manuscript as this transect crosses a segment of the margin where sill sheets have been emplaced through under- (and mid-) accretion (as highlighted in the discussion), and therefore development of these sill-tip faults is inhibited by the overlying sill sheets, resulting in a monoclinal, rather than stepped margin.

(2) Fully agree, porosity reduction within the host rock (dominantly by compaction, cataclasis and the formation of deformation bands) will accommodate a percentage of additional magma volume, so surface analysis alone will underestimate the total volume of magma emplaced.

(3) Cross-cutting relationships of brittle structures are very challenging and often difficult to discern. As highlighted above, the deformation here is dominantly via conjugate shear fractures/ deformation bands, and so even within a single deformation pulse we will naturally see cross-cutting structures. Y-nodes may, therefore, represent a cross-cutting fracture, but they can just as easily represent a fracture termination against a

pre-existing structure.

In our comparison of a forced fold above a normal fault versus a series of stacked sill sheets (Fig. 9), we propose that deformation is likely to be progressive (equivalent to multiple slip events on a normal fault as it grows) and so identification of discrete events will be challenging. Without additional fracture characteristics (such as mineral precipitates associated with individual sheets) it will be very difficult to differentiate discrete events.

Having said this, this is something that will certainly be worth further investigation.

---

## Referee Comment (RC2) · Laurel Goodwin (Referee) · 18 Aug 2020

General Comments

This beautifully illustrated paper is divided into two parts: (1) a description of deformation band networks associated with the Trachyte Mesa intrusion and analysis of their physical significance and (2) an evaluation of the impact of these networks on fluid flow. This overview addresses each part, then considers the paper as a whole. Specific comments re: figures and text follow.

[Figure]

The core of this manuscript is a description of the orientations and detailed analysis of the topology of deformation band networks that are demonstrably physically associated with the Trachyte Mesa intrusion. The methods used, results presented, and limitations of data provided in this section of the paper are remarkably clear, honest, and well documented. I was particularly pleased to see a specific, clear, and quantifiable description of terms such as 'intensity' that are commonly used loosely to mean several different things. The authors note that quantitative assessment of network topology was undertaken only on surfaces oriented at a high angle to the average strike of deformation bands documented by previous studies – a reasonable choice of approach to maximizing information acquired given likely constraints on time and outcrop. The results of this work are compelling. In my opinion, they represent the most significant contribution of the manuscript, showing that deformation bands increase in abundance, length, and connectivity across the margin of the intrusion. I agree with the authors that quantitative studies such as this "are essential to improve and constrain laboratory and numerical models of intrusion emplacement mechanisms and associated deformation." The only substantive criticism I have of this portion of the manuscript is the use of the term 'fracture' to refer to deformation bands. I know that small-displacement fractures and deformation bands are sometimes lumped together but disagree that such lumping is constructive. A fracture is a surface across which cohesion has been lost; the same is not true of a deformation band, which is a zone of deformation within which fractures may or may not form. I therefore suggest modifying the text to state that topological analysis of fractures can be extended to other discrete structures and in this case has been applied to deformation bands. Subsequent references to the structures studied should consistently refer to them as deformation bands.

In addition to the detailed analysis of attributes of the deformation band network, the authors include a qualitative assessment of porosity and permeability of the deformation bands. In contrast to the evaluation of geometry of deformation band networks, this analysis is neither rigorous nor well documented. Porosity was estimated for host rock in one thin section from each field station and for deformation bands in thin sections

from the four stations that included them. Estimates were made by visual comparison with charts in a sedimentary petrology textbook. Data are given in a plot (Fig. 7), with a single value for deformation band porosity and a range of values for host rock porosity. We are not told what magnification was used for analyzed images, how many images were analyzed (does the fact that a range of porosity is not reported for deformation bands indicate that only one image was used to evaluate deformation band porosity in each sample, or that porosity is surprisingly constant in deformation bands?), or what size area was analyzed for each sample. We are not told if the authors took the time to determine an appropriate representative elemental area for these samples. [Porosity will vary within a sample; however, the average porosity of multiple measurements will level off when a representative elemental area is attained.] The use of visual comparison charts is also an issue. In my experience with students using such charts in petrography labs, this qualitative analysis of the same area by different individuals can vary as much as 10%. This is not an accurate approach to analyzing porosity. In my opinion, a more rigorous analysis is necessary to attain publishable data. Point counts are always an option. I suspect, however, that it would be faster and more accurate to download ImageJ, analytical freeware that is commonly used for this purpose (in addition to analysis of such variables as percent cement or grain size distribution), from this site: https://imagej.nih.gov/ij/download.html. Image processing to (for example) separate out blue areas that represent pores can facilitate analysis.

The approach the authors used to infer permeability reduction associated with deformation band formation – "assuming Kozeny-Carmen equation fundamentals" (line 341) is not clear. The equation was developed to estimate permeability in sediments based not only on porosity, but also on grain size and sphericity (both of which change in cataclastic deformation bands). How were they able to extract relative permeability by 'assuming' these 'fundamentals'?

In sum, the authors present a beautifully illustrated and detailed quantitative analysis of deformation band networks observed in a transect across the margin of an igneous

body intruded as a series of sills. They then present an intriguing qualitative analysis of the hydrologic significance of these structures. Data collection, analysis, and interpretations in the latter portion of the paper require some revision for accuracy, clarity, and rigor, after which I recommend publication.

Specific Comments Linked to Figures

The authors indicate that the outcrops studied are all part of "massive" sandstone roughly 10 m thick, implying that the samples collected are all part of a single host rock unit. The term "massive", however, is applied by sedimentologists to strata that are structureless, either from the time of deposition or due to post-depositional processes such as bioturbation. However, it is evident from the images and descriptions of sedimentary features provided by the authors that the outcrops studied are neither structureless nor uniform. Figure 2 beautifully illustrates both lateral and vertical variations in sedimentary structures, as well as subtle differences in color and resistance to weathering, consistent with variations in grain size and/or cement mineralogy or percent. In addition to removing the term 'massive' from the paper, I propose the authors explicitly state that although it is not possible to trace a single bed across the margin of the intrusion, their analyses suggest they have sampled rocks with relatively similar grain size, grain rounding, and mineralogy.

Figure 8 is very attractive, but not designed for ease of understanding. I'm a microstructure geek, and I found it hard to navigate because part of the information that would normally be provided in the caption of a single image is given in the text, some is in the caption, and some is beneath a single figure. Some of the labels on images are very difficult to see. For example, I searched for Fe labels after I saw in the caption that Fe referred to 'iron staining' (staining of what? does this mean iron oxide grains or cement or coating?). The dark text does not show up on dark background. Red labels are hard to see; DB labels should be backed with white boxes to stand out and arrows generally need to be larger (only the TMFS-6 arrows really stand out). In general, it would be better if labels were bold; those imposed on dark areas of thin sections should be white. In

short, it is not possible to glean all of the important information about an image from the figure and caption alone. Because the data acquired from thin sections are important to this story, I suggest a different approach. Move the partial captions beneath each image into the main caption and add information. For example: TMFS-1 (∼20 porosity), TMFS-2 (∼15-20% porosity), and TMFS-3 (∼30-35% porosity) are all well sorted, subrounded, subarkoses with local poikilotopic calcite cement. Only TMFS-3 includes deformation bands. Porosity is reduced to <5% in the deformation band, within which small, angular grain fragments provide evidence of limited cataclasis (example highlighted with a bold arrow). Walk the reader through the rest of the photomicrographs in a similar way. Be sure to clearly state what you see as well as what you infer. You don't see pressure solution; you infer it from embayments in grains at point contacts (which can be better highlighted with bold arrows). You don't see cataclasis, you infer it from angular grain fragments. You don't see compaction; you infer it from reduced porosity and preferred alignment of elongate grains (which you don't mention anywhere, but should). In other photomicrographs you can see alignment of elongate clasts parallel to cross laminae or deformation bands. It's good to point that out. Also, I personally like the fact that you haven't drawn lines over deformation band boundaries. For readers less familiar with what these features look like, you may wish to provide some guidance in either words or arrows that mark top and bottom boundaries to a band.

You note 'indistinct "fuzzy" boundaries to larger grains' beneath your last photomicrograph. Most of the grains are quartz and have sharp margins. Your labeled plagioclase grain has "fuzzy" margins, which are also locally brownish in color. Without being able to zoom in further or look at this on an SEM, I would say that there are several things that could contribute to this appearance. Top on my list is margins that are oblique, rather than perpendicular, to the surface of the thin section. Where the edge of a grain dips away from the grain center, it will be increasingly out of focus with distance. With extensive cataclasis, you may be looking through a zone of fine grain fragments on that grain edge. I think this is what you are referring to, but I'm not sure. If it is, spell it out and highlight the specific margin. If I were you, however, I would focus on more obvious evidence of cataclasis: a high percentage of angular grains that are substantially reduced in size with respect to subrounded grains evident in host rock.

On p. 8, you also discuss 'early development of sub-grain boundaries', and follow that on p. 9 with observations of 'clear sub-grain boundaries parallel to deformation band orientations.' In general, we use the term 'subgrain' (with no hyphen) to refer to a part of a larger grain separated from the host grain by a dislocation wall. Production of subgrains is part of the process of rotation recrystallization; it is not a brittle process. Subgrain boundaries are only visible with crossed polars, which causes differences in orientation of the crystal across these dislocation walls to show up as differences in exteniction (grayscale). The only features I see oriented subparallel to deformation bands appear to be cracks. Please revise the text for clarity and accuracy.

Specific Comments re: Text and Interpretations

Add a reference to the list of studies of deformation band impacts on flow (line 40): Sigda, J.M., Goodwin, L.B., Mozley, P.S., and Wilson, J.L., 1999, Permeability alteration in small-displacement faults in poorly lithified sediments: Rio Grande rift, central New Mexico: In Haneberg, W.C., Mozley, P.S., Moore, J.C., and Goodwin, L.B. (eds) Faults and Subsurface Fluid Flow in the Shallow Crust, AGU Monograph 113, 51-68.

Change lines 51-53: "Deformation bands preferentially form in more poorly lithified layers within quartz arenite to arkosic sandstones (i.e. those lacking in lithics) at shallow depths (1–3 km; Fossen, 2010)" to: "Deformation bands within quartz arenite to arkosic sandstones (i.e. those lacking in lithics) preferentially form in more poorly lithified layers at shallow depths (1–3 km; Fossen, 2010)." The former suggests deformation bands are restricted to poorly lithified layers of specific composition.

In lines 195-196, the authors refer to 'cycles enclosing blocks' and note common features of 'networks with lots of cycles'. The discussion of cycles refers to Figs. 6c and 2b&c, but it is not possible to understand how the reader is supposed to connect this information to the images. The term 'cycle' is not defined, and it is never mentioned

again. If it is important to the story, the authors should define what they mean and why it is relevant. If it is not, they should remove references to 'cycles'.

In line 218, the authors refer to 'a slightly coarser grained bed within the sandstone horizon'. I am not aware of a definition of 'horizon' used in this context. It appears to be a way to suggest associations between samples collected. Does it refer to the 4 m thick section of sandstone shown in Fig. 2? Please clarify.

I would like to see the authors replace references to 'weak' deformation or cataclasis with more specific information regarding observations rather than interpretations. I suspect they mean that evidence of fracture and associated grain-size and porosity reduction is present, but not as extensive as in other samples, as suggested by higher estimates of porosity.

I suggest the authors replace 'grain crushing' with 'distributed microcracking' in places like line 233. I think it is a more accurate representation of the variable amounts of grain-size reduction via fracture illustrated in their thin sections. Their photos show a range from deformation bands in which the majority of grains are subrounded and similar in size to those in the host rock to deformation bands in which most of the grains have been reduced to relatively small angular fragments and relatively few original grains remain.

On line 235, replace 'Calcite is also present' with 'Calcite locally fills pores'.

Line 245 refers to early development of subgrain boundaries. I addressed misconceptions re: subgrain boundaries in the previous section on Specific Comments Linked to Figures. The authors should make appropriate changes to the text here also.

On line 248, the authors discuss embayed contacts. I think it would be helpful to clarify what is meant by 'embayed', with reference to more clearly annotated examples in thin section images.

The sentence beginning on line 256 states that 'Haematite is also incorporated into

the matrix within deformation bands as a result of quartz grain crushing. Note the brownish-staining of deformation bands in Figs. 7a and 8". What evidence supports this interpretation? Is it possible that hematite was precipitated after formation of deformation bands? Please provide evidence (. . .and you don't need to hyphenate brownish & staining).

Line 305 refers to 'minor cataclasis as evidence for shear'. Minor cataclasis can occur by compaction alone. It doesn't require shear.

On line 309, the authors propose that evidence of compaction in sandstone suggests confining pressure may increase with proximity to the intrusion. It is certainly a sign of shortening, consistent with intrusion, but that suggests an increase in margin-perpendicular stress, not an increase in confining pressure. Note also that intrusions, particularly shallow crustal intrusions, cool very rapidly. The temperature gradient between thin sheets of partially crystalline magma and wall rock so shallow it still has high porosity is very high, and temperature dissipates rapidly at cool shallow temperatures. If you know the thickness of individual sills and likely depth of intrusion, you can do a back of the envelope calculation to determine the cooling rate for a normal geothermal gradient (or even a slightly elevated one, which would not produce high temperatures at relatively shallow depths where you see high porosity sandstones). 'Pressure solution' actually has nothing to do with pressure. It is caused by a stress-induced chemical potential gradient. I suspect that what you are seeing is that the deformation bands that have accommodated the greatest deformation have the highest number of high-stress point contacts between grains, where solution mass transfer is facilitated.

Line 311: Crush breccias do not consist of fragments that are only visible with a microscope. You do show clear evidence of cataclasis, which could be defined as distributed microcracking and rotation and translation of resulting clasts. You might want to provide a definition like this where you first introduce the term in the paper, to facilitate discussion here.

Lines 313-314: If a principal slip surface or fault core were present, you would call it a fault or a deformation band fault and not a deformation band.

Line 317: I am not familiar with the term 'permeability pathway'. Deformation bands are features that can influence flow pathways, but they cannot be considered in isolation. In this case, the elephant in the hydrologic room is the extremely low permeability intrusion. Regional flow will take the 'easiest' route around the intrusion, which will be influenced not just by deformation band distribution and connectivity but also by the permeability of the surrounding undeformed rocks. This paragraph also reflects a misunderstanding of the hydrologic significance of microstructural observations. The fundamental misapprehension is that tabular structures that formed by different processes (e.g., compaction bands vs. cataclastic deformation bands) can influence flow differently even if they have the same permeability.

Your intrusion is effectively an impermeable wall. Your deformation band networks may redirect or inhibit flow in a shell around the plutons, or the main effect may be created by the intrusion itself. The best way to determine these effects would be to measure the permeabilities of cores cut in different directions through the networks, then work with a hydrogeologist to model flow. Without those data, I think you are restricted to providing a clear description of the structures at different scales. Please appreciate that the description itself is a significant contribution.

Lines 327-328: I do not know if anyone has published evidence of magmatic fluids of appropriate composition to precipitate calcite. I think this would be a more compelling suggestion if the authors could cite a study indicating it was possible. I do know that the solubility of calcite decreases with increasing temperature, so I suspect that heat introduced by the intrusion could facilitate precipitation of calcite from surrounding groundwater of appropriate composition.

Lines 335-336: I think it is particularly important to replace 'fractures' with 'deformation bands' in these sentences.

Line 342: I don't know what the authors mean by "However, this assumes a homogeneous development of the grain-scale processes". Please explain.

In line 351, the authors state "At Trachyte Mesa, deformation bands decrease markedly from ~5 to 10m above the intrusion margin.." I assume they mean deformation band intensity decreases. However, they do not present data showing vertical variations in deformation band networks. Is this a personal observation? If the authors have data that show this variation, they should provide it. If relevant data have been published, they should cite the reference.

I suggest the authors modify lines 354-357 to state: "In addition to reducing the bulk permeability of the reservoir, the deformation bands largely strike parallel to the intrusion margin (Wilson et al., 2016), producing an anisotropy in permeability similar to that of a fault zone (e.g. Farrell et al., 2014).

Line 360-361 should be modified to state: 'Gaining a better understanding of these emplacement-related deformation structures may have important implications for fluid flow, hydrocarbon reservoir connectivity / deliverability, hydrology, geothermal energy and $CO_2$ sequestration...'

I suggest the authors modify line 378-379 as follows: "The increase in margin-parallel Y- and X-nodes with proximity to the intrusion is likely to inhibit flow perpendicular to the intrusion margin, as well as potentially forming non-producible reservoir zones."
* * *

---

## Author Response (AR1)

[revised manuscript text omitted]

**Comment [P5]:** Updated fig d with larger sample station locality symbols and increased font size

**Figure 1: Geological setting and study area. (a) Simplified maps showing location of the Henry Mountains intrusive complex and the Trachyte Mesa intrusion. Intrusion  areas adapted from Morgan et al. (2008). (b) Contoured (20 m intervals) aerial image (Source data: National Agricultural Imagery Program, https://gis.utah.gov/data/aerial-photography/) of the Trachyte Mesa area showing the intrusion outline (yellow) and study area. Dashed lines in the SW depict the sub-surface extent of the intrusion, as defined by Wetmore et al. (2009) from magnetic resistivity data. Blue dots show field localities visited as part of wider reconnaissance studies (Wilson et al., 2016). (c) Satellite photograph (© Google Earth) of the study area (NW margin of the Trachyte Mesa intrusion). Structural stations for fracture studies indicated by numbered red dots. Yellow dashed lines show outcrop exposure of sill sheet terminations. (d) Field photograph showing monoclinal geometry of the NW intrusion margin. Note blocky, red Entrada Sandstone units concordant with the underlying intrusion top surface, and stacked intrusive sheets below (sheet terminations highlighter in yellow). Structural Stations 2 – 6 are indicated (red dots). Viewpoint location shown in (c). Contoured equal-area stereoplot shows poles to planes for deformation bands measured across the NW-margin of the Trachyte Mesa intrusion (from Wilson et al., 2016).**

[Figure]

**Figure 2: Sampling traverse across lateral margin of the Trachyte Mesa intrusion. (a) Panoramic photograph of study area. Note red, cross-bedded Entrada Sandstone unit, and Trachyte Mesa intrusion outcropping to the right. All structural stations lie within the same  cross-bedded sandstone unit. Although this unit exhibits lateral and vertical variations in sedimentary**

**structures, attempts were made to sample rocks with  similar grain size, grain rounding, and mineralogy**
680 **for this study.** (b) – (g) Overview outcrop photographs for each station.

[Figure]

**Figure 3: Fracture analysis of hand specimens. (a) – (f) Bedding-normal cut surfaces with hand-specimen photographs showing fracture analysis circles for each structural station (note, sample numbers correspond to their respective station). Fracture analysis was carried out on freshly cut surfaces. Circular scans show the fracture network and associated I-, Y-, X- and E-nodes (for explanation of terminology see Fig. 4 and Sanderson and Nixon, 2015). Statistics show total number of branches (N), total fracture length (SL), fracture density/ intensity ($P_{21}$), and branch dimensionless intensity ($B_{22}$).**

[Figure]

Figure 4: Schematic image outlining the principal method applied for fracture analysis (Sanderson and Nixon, 2015). Branches and nodes are shown on fracture trace (A–B): I-nodes (red circles); Y-nodes (blue triangles); X-nodes (green squares). Proportions of I-, Y- and X-nodes may be plotted on a ternary plot to visualise different fracture network types (after Manzocchi, 2002).

690

[Figure]

**Figure 5: Example of fracture analyses undertaken at different scales in this study, sample TMFS-5. (a) Outcrop photograph with superimposed fracture analysis circle showing the fracture network and associated I-, Y-, X- and E-nodes. (b) Hand specimen analysis (see Fig. 3). (c) Whole thin-section photograph (taken using flatbed scanner) and plain-polarised light (PPL) photomicrograph. Optical microscopy petrographical, porosity and microstructural analyses were carried out on thin sections cut from each hand specimen. Sections were impregnated with blue resin to highlight porosity.**

[Figure]

**Comment [P6]:** New Figure to show example of the ImageJ porosity analysis workflow

**Figure 6: Quantitative determination of porosity percentages using ImageJ. (a) Sections were impregnated with blue-dyed plastic resin in order to highlight porosity. (b) Image analysis using ImageJ to select areas (red) containing blue dye. (c) Percentage areas (porosities) can then be calculated for both whole section and selected areas (e.g. host rock vs deformation band). Note the substantially lower porosity and smaller pore size within the deformation band.**

[Figure]

**Comment [P7]:** Added B22 values for structural stations in Fig c as per reviewer 1's recommendation

**Figure 67:** Nodal and fracture analysis results. (a) Schematic diagram showing relative location of each structural station across the monoclinal intrusion margin. (b) Bar chart showing spatial variation in nodal populations. (c) Triangular plot showing ratio of I-, Y- and X-nodes for total values for each station. Contours represent the branch dimensionless intensity (B$_{22}$) of a network with that nodal topology when it is at the percolation threshold (i.e. the limit or threshold at which the network is "unconnected"/ "connected"). If the network has a higher B$_{22}$ than in the contour plot then the network is considered connected; conversely if lower it is considered unconnected. As the B$_{22}$ values for Stations 1 and 2 (see Table 1 and values in key) are below the contour values, these deformation band networks are clearly unconnected, whereas Stations 4, 5 and 6 are well above the contour values and thus may be considered well connected. For more details on the triplot template, see Sanderson and Nixon (2018). (d)-(f) Summary plots showing the log of log-linear plots showing various fracture attributes at each station (see Table 1 for values). <L>: Mean line length (m; total line length/ number of lines); P$_{20}$: line frequency; P$_{21}$ (m$^{-2}$; number of lines/ sample area): line intensity (m$^{-1}$; total line length/ sample area); P$_{22}$: line dimensionless intensity (multiplying line intensity by the mean length); : Mean branch length (m; total branch length/ number of branches); B$_{20}$: branch frequency (m$^{-2}$; number of branches/ sample area); B$_{21}$: branch intensity (m$^{-1}$; total branch length/ area); B$_{22}$: branch dimensionless intensity (multiplying branch intensity by the mean length); <d>: average degree of nodes (the number of branches that meet at a node); C$_L$: connections per line; C$_B$: connections per branch.

[Figure]

(a)

(b)

**Comment [P8]:** Updated Fig b with porosity values derived from ImagJ analsysis. Note, absolute values have decreased for all samples using ImageJ method, but relative trends are still the s as before

**Figure 78: Porosity variability across the intrusion margin. (a) Whole thin-section photographs (flatbed scans) for each structural station. Blue dye denotes porosity in each sample. (b) Plot showing variability in porosity observed in this study for each station. Porosity percentages were calculate using the image analysis software package ImageJ as shown in Fig. 6. estimated using visual comparison charts (Bacelle and Bosellini, 1965, Tucker, 2001). Note, a similar porosity reduction trend has been observed previously (see fig. 8 in Morgan et al., 2008). DB: Deformation Band; X-lam: Cross-lamination.**

725

[Figure]

**Figure 89:** **Thin section photomicrographs** **host rock composition structure and porosity**. **Note overall decrease in porosity (blue dye staining) and increase in deformation**  **from sample TMFS-1 through to TMFS-65. (a)** **Undeformed host rock sample TMFS-1 viewed under plane polarized light (ppl) showing well-sorted subarkose host rock, variable porosity ~9.4–13.8 % due to patchy calcite spar (note large poikilotopic calcite spar in centre of section); (b) Same section as (a) viewed under cross polarized light (xpl); (c)** **Undeformed host rock sample TMFS-2 (viewed in** **xppl) showing a well-sorted subarkose host rock, total porosity ~12 %, and patchy calcite spar; S(d)** **High porosity zone (23.4 %) in sample TMFS-3 (ppl); (e)** **Deformed host rock sample TMSF-4 (ppl), 16.4 % porosity, note the pressure solution, embayed contacts and intragranular microcracks and microfractures (both tensile and shear fractures observed) in multiple quartz grains (see red arrows for examples); (f)** **Deformed host rock sample TMFS-5 (ppl) showing markedly reduced porosity (9.1 %), further deformation structures (intragranular fractures, embayed contacts and pressure solution), apparent grain size reduction and tighter pack in of grains. Cal: Calcite spar; DB: Deformation Band; Fe: Iron staining; Pl: Plagioclase Feldspar; Qtz: Quartz. Red arrows highlight zones of pressure solution, embayed contacts, and microfractures. Porosity values from ImageJ image analysis.**

Cal: Calcite spar; DB: Deformation Band; Fe: Iron staining; Pl: Plagioclase Feldspar; Qtz: Quartz. Red arrows highlight zones of pressure solution, embayed contacts, and grain shear.

[Figure]

**Comment [P11]:** New Figure to repl
previous montage. Larger images, additi
of XPL photos, and enhanced labels as p
Reviewer 2 comments

745

**Figure 10:** Thin section photomicrographs showing deformation band structure and porosity (edges of deformation bands highlighted in by red markers at edges of image). (a) Poorly developed deformation band in sample TMFS-3 (viewed in ppl) showing minor cataclasis (microcracks and fractures), embayed contacts and an increase in calcite cementation within the deformation band. Note the marked decrease in porosity within the host rock (13-23 %) and the deformation band (2.8 %); (b) Deformation band in sample TMFS-4 (viewed in ppl) showing intragranular microcracks and transgranular fractures (red arrow), grain size reduction, and an increase in calcite cementation within the deformation band (host rock porosity 12.9-14.9 %, deformation band porosity 1.7 %; Fig. 6); (c) and (d) Deformation band in sample TMFS-5 viewed under ppl (c) and xpl (d), (host rock porosity 7.4 %,  def. band porosity ~1.1 %).  Cataclastic fabrics within the deformation band includes angular smaller grains (grain size reduction), intragranular microcracks and microfractures, and increase in calcite cementation within the deformation band. Note, in xpl (d) the subgrain boundary within large quartz grain sub-parallel to band orientation, as well as microcracks, transgranular fractures and undulose extinction in some quartz grains outside the defined deformation band; (e) and (f) Deformation band in sample TMFS-6 viewed under ppl (c) and xpl (d), (host rock porosity 3.2-4.7 %, deformation band porosity ~1.6 %).  Deformation band shows well defined cataclastic fabrics (angular smaller grains, intragranular microcracks and microfractures, sheared and rotated grains) and an increase in calcite cementation. Note, the many strained grains (microcracks, transgranular fractures, undulose extinction, rotated grains and embayed contacts) outside the defined deformation band; (g) Sample TMFS-6 (viewed in ppl) showing deformation band cross-cutting a cross lamination within the sandstone (see annotation showing orientation of DB relative to laminae). Note the dominance of small angular grains in the section, reflecting grain fracturing within deformation band. (h) Sample TMFS-6 (viewed in ppl) showing microporosity with deformation band, note clean angular contacts to quartz grains and less distinct ("fuzzy") grain boundaries to feldspar grains. Angular shear fracture in quartz grain highlighted with red arrows. Cal: Calcite spar; DB: Deformation Band; Fe: Iron staining; Pl: Plagioclase Feldspar; Qtz: Quartz; X-Lam: cross-lamination. Red arrows highlight zones of pressure solution, embayed contacts, and microfractures. Porosity values from ImageJ image analysis.

**Comment [P15]:** Updated Fig b to show moderate deformation above intrus as per Reviewer 1's comments and additional text added

**(a) Forced fold above normal fault**

[Figure]

| Zone | 1 | 2 | 3 | 4 | 5 | 6 | 7 | 8 | 9 |
|---|---|---|---|---|---|---|---|---|---|
| $e_x$ | -5 | -2.5 | -7.5 | -2.5 | -10 | -22.5 | -35 | -2 | -5 |
| $e_y$ | 0 | +2.7 | +2.7 | +3 | +1.6 | +6 | +5.3 | +4.5 | +1.3 |
| $e_z$ | 0 | 0 | 0 | +7.5 | +7.5 | +15 | +35 | +5 | 0 |

Deformation bands appear to decrease markedly above intusion (though outcrop exposure is limited)

**(b) Monoclinal intrusion margin**

Increase in deformation band frequency, intensity & dominance of Y-nodes

Stacked sill sheets

**(c)**

'Out-of-sequence' sill stacking (under- & mid-accretion)

Sill 1
Sill 3
Sill 2

Normal sill stacking (over-accretion)

Sill 2
Sill 1

770

**Figure 911: Schematic 3D block diagrams and cross sections comparing the distribution of deformation structures. (a) A forced fold above a normal fault (modified after Ameen, 1990 and Cosgrove, 2015). (b) Deformation bands across the Trachyte Mesa intrusion (this study). (c) Cartoon showing varying deformation styles and distribution in relation to the order of sill sheet stacking (Wilson et al., 2016). In (a) the fold is divided into zones (see inset table) depending on the level of strain normal ($e_z$) and parallel to the layer ($e_y$ parallel to the fold hinge and $e_x$ normal to it). Note: extension is negative and contraction positive. Coloured zones highlighted in (b) are solely for visual purposes and do not correspond to the strain zones defined in (a).**

| | | | TMFS_1 | TMFS_2 | TMFS_3 | TMFS_4 | TMFS_5 | TMFS_6 | *units* |
|---|---|---|---|---|---|---|---|---|---|
| | | | 1 | 2 | 3 | 4 | 5 | 6 | |
| **Nodes** | I-nodes | I | 11 | 4 | 15 | 48 | 102 | 2 | |
| | Y-nodes | Y | 9 | 3 | 44 | 83 | 528 | 435 | |
| | X-nodes | X | 4 | 0 | 22 | 19 | 217 | 114 | |
| | # Nodes | \|N\| | 24 | 7 | 81 | 150 | 847 | 551 | |
| | | | | | | | | | |
| **Topology** | # Branches | \|B\| | 27 | 6.5 | 117.5 | 186.5 | 1277 | 881.5 | |
| | # Lines | \|L\| | 10 | 3.5 | 29.5 | 65.5 | 315 | 218.5 | |
| | Average degree | <d> | 2.25 | 1.86 | 2.90 | 2.49 | 3.02 | 3.20 | |
| | Proportion (Y+X) | P(x+y) | 0.70 | 0.43 | 0.82 | 0.70 | 0.91 | 1.00 | |
| | | | | | | | | | |
| **Line** | Frequency | P20 | 23.55 | 3.59 | 57.00 | 157.0 | 461.5 | 843.0 | *m-2* |
| | Intensity | P21 | 9.79 | 3.77 | 13.40 | 26.72 | 49.07 | 80.87 | *m-1* |
| | Dimensionless intensity | P22 | 4.07 | 3.96 | 3.15 | 4.55 | 5.22 | 7.76 | |
| | Ave. Line length | <L> | 0.42 | 1.05 | 0.24 | 0.17 | 0.11 | 0.10 | *m* |
| | Connections per line | $C_L$ | 3.70 | 1.71 | 4.44 | 3.19 | 4.93 | 5.03 | |
| | | | | | | | | | |
| **Branch** | Frequency | B20 | 82.0 | 6.7 | 225.0 | 453.3 | 1932.8 | 3401.1 | *m-2* |
| | Intensity | B21 | 9.79 | 3.77 | 13.40 | 26.72 | 49.07 | 80.87 | *m-1* |
| | Dimensionless intensity | B22 | 1.17 | 2.13 | 0.80 | 1.57 | 1.25 | 1.92 | |
| | Ave. Branch length |  | 0.12 | 0.56 | 0.06 | 0.06 | 0.03 | 0.02 | *m* |
| | Connections per branch | $C_B$ | 1.78 | 1.38 | 1.87 | 1.76 | 1.94 | 2.00 | |
| | | | | | | | | | |
| | Proportion of I-nodes | P(I) | 0.46 | 0.57 | 0.19 | 0.32 | 0.12 | 0.00 | |
| | Proportion of Y-nodes | P(Y) | 0.38 | 0.43 | 0.54 | 0.55 | 0.62 | 0.79 | |
| | Proportion of X-nodes | P(X) | 0.17 | 0.00 | 0.27 | 0.13 | 0.26 | 0.21 | |

**Table 1: Summary table showing total values for each structural station. Total values do not include hand specimens due to their small size making estimates of frequency and intensity unreliable, though trends for samples do match those for outcrops in this study. For details on individual scan circles see supplementary materials.**

Dear Prof. Eichhubl

**Response to the Editor post reviewer comments; submission MS No.: se-2020-71**

We have done our best to implement all recommendations and changes made by the two reviewers. For line by line specific changes relating to reviewer comments, please see the replies to each reviewer below.

Below are the three main technical updates following reviewer feedback.

**1) Fractures vs Deformation bands.**

In accordance with the steer from both yourself and Laurel we have replaced all reference to "fractures" with "deformation bands". The only references to fractures now is with regards a general introduction to the established methodology and we then make it clear that we are extending this methodology here to the study of deformation bands (lines 72-73 in tracked changes document).

**2) Porosity analysis**

We agree entirely with Reviewer 2 (Laurel Goodwin) that the porosity analysis method would be improved by using image analysis software; this was something we had discussed as an author team during the original manuscript preparation. We have therefore updated the porosity studies sections (4.2. Microstructural Analysis, lines 217-277) with results of porosity analysis using ImageJ. We have added a new image (Fig. 6; line 700) to show an example of the ImageJ analysis workflow, and have also updated the graph in Figure 8b (line 722). Note the absolute porosity values are lower using this ImageJ workflow compared to those derived from visual analysis; however, the relative numbers and trends are consistent across both methods.

**3) Microstructures terminology and images**

We have reviewed and revised various microstructural descriptions in accordance with feedback from Reviewer 2. To improve the image resolution and details presented in the Microstructures figure (formerly Fig. 8), we have split this figure into two new photo montages (now Figures 9 and 10; lines 730 and 745) to show host rock and deformation band microstructures respectively.

We believe these changes and updates make for a much improved manuscript. We are greatly appreciative of each reviewer's feedback and think that the edits make the paper an even stronger contribution for your journal volume.

Yours sincerely,

Dr Penelope Wilson

**Reviewer 1 Response:**

Dear Reviewer 1

Many thanks for your positive and constructive comments on the manuscript. We've tried to address all items you have raised, and think that the updates make for a much improved manuscript

Please find below our responses to the various points you raise. We have a word document with tracked changes, should you wish to see it.

Kind regards,

Dr Penelope Wilson

Review Comment – Specific Comments: One specific area that I would like to see developed a bit further is the conceptual model (Fig 9) presented in the final discussion (Sections 5.1 and 5.2). You compare the patterns of deformation you observe at Trachyte Mesa (9b) to those in a forced fold above a normal fault (9a). I hope you can expand this discussion to address how differences between the processes might be reflected in the patterns of deformation observed. The sills have intruded laterally underneath the entire mesa (i.e. see Fig. 12 in Morgan et al., 2008), whereas the forced fold formed over either an upward or laterally propagating fault tip (e.g. White and Crider, 2006). One might therefore expect a structural density greater than the background above the intrusion but not above the footwall of the normal fault. Although your sample transect doesn't appear to extend far enough to directly address this question, it would be nice if you speculated on it a bit in the discussion. Perhaps there might also be differences in the orientations of structures?

Author Response – Unfortunately there are limited host rock exposures overlying the intrusion top surface, as ideally we would have liked to have extended the transect onto the top surface as you suggest. We agree in order to accommodate the additional volume of magma we would expect to see some form of compaction and/or deformation in the host rocks above the intrusion. The few outcrops we did find during our wider field studies did not appear to exhibit significant deformation band structures; however, the host rock did appear to exhibit reduced porosity in outcrop (either compaction of thermal). When studying the neighbouring intrusion host rock outcrops at Maiden Creek, host rock outcrops did show evidence for stylolite development. We may, therefore, speculate that these may also have been present above Trachyte Mesa.

We have added some extra text in the paper to address the points you have raised, and have also modified the final Figure to highlight some potential deformation above the intrusion top surface.

Review Comment – I also found it a bit confusing that you mention that Sanderson and Nixon (2015) argue for use of branch attributes (vs length) when characterizing fracture networks (lines 142-143), but then proceeded to focus on length attributes in your discussion of the results (lines 186-189). Perhaps you should drop this point from the earlier discussion or else recast your results to emphasize branch attributes?

Author Response – Agree, we have reworded the results section so that we refer predominantly to branch data in the first instance.

Review Comment – Other Minor/Technical Corrections:

Review Comment – Line 45 an 'n' is missing from Sternlof

Author Response – good spot, corrected

Review Comment – Line 56 'implication for' rather than 'on'

Author Response – Done

Review Comment – Lines 114-116. Please add some estimate of the average (central tendency) orientation of deformation bands and analysis sections here.

Author Response – We've included a contoured stereonet in Figure 1d show the orientation of deformation bands is predominantly perpendicular to the transect orientation and cite the Wilson et al., 2016 for further orientation analysis. Hopefully this will suffice.

Review Comment – Fig 1 caption. Change 'outlines' to 'areas'.

Author Response – Done

Review Comment – Fig. 6c. It would be nice to have the B22 values for each point annotated somewhere on the plot, either in the legend or with the point labels.

Author Response – Good idea, we've added B22 values in the key in Fig. 6c to aid the reader in comparing locations in the trip plot against this locality attribute.

Review Comment – Figure 6 caption: Instead of using "log-linear", which implies that the station numbers have quantitative meaning, reword to "Summary plots showing the log of various fracture attributes at each station"

Author Response – Done.

**Reviewer 2 Response:**

Dear Laurel

Many thanks for your detailed and highly constructive comments on the manuscript. We've tried to address all items you have raised, and think that the updates make for a much improved manuscript.

Following your recommendation, we have replaced the visual qualitative porosity estimates with more quantitative values derived using ImageJ. This had been part of our original work plan, but did not have time to do this prior to our original submission! We've added a figure highlighting the basic workflow used in ImageJ, updated the graph in Figure 8 (formerly 7) and all porosity values within the text to refer to these new values. Note, all porosities are markedly lower using this image analysis method, but the overall trends are consistent with earlier observations.

With regards the reference to "Kozeny-Carmen equation fundamentals", this is simply a discussion point and we have not actually attempted to estimate permeabilities ourselves. We've therefore added a few more words to highlight that applying this equation to deformation band permeabilties is a gross over-simplification.

Below are a list of the point by point remarks you raised and the actions we have taken.

Kind regards,

Dr Penelope Wilson

Reviewer Comments – The authors indicate that the outcrops studied are all part of "massive" sandstone roughly 10 m thick, implying that the samples collected are all part of a single host rock unit. The term "massive", however, is applied by sedimentologists to strata that are structureless, either from the time of deposition or due to post-depositional processes such as bioturbation. However, it is evident from the images and descriptions of sedimentary features provided by the authors that the outcrops studied are neither structureless nor uniform. Figure 2 beautifully illustrates both lateral and vertical variations in sedimentary structures, as well as subtle differences in color and resistance to weathering, consistent with variations in grain size and/or cement mineralogy or percent. In addition to removing the term 'massive' from the paper, I propose the authors explicitly state that although it is not possible to trace a single bed across the margin of the intrusion, their analyses suggest they have sampled rocks with relatively similar grain size, grain rounding, and mineralogy.

Author Response – Removed the term "Massive" and added additional wording as proposed above.

Reviewer Comments – Figure 8 is very attractive, but not designed for ease of understanding. I'm a microstructure geek, and I found it hard to navigate because part of the information that would normally be provided in the caption of a single image is given in the text, some is in the caption, and some is beneath a single figure. Some of the labels on images are very difficult to see. For example, I searched for Fe labels after I saw in the caption that Fe referred to 'iron staining' (staining of what?

does this mean iron oxide grains or cement or coating?). The dark text does not show up on dark background. Red labels are hard to see; DB labels should be backed with white boxes to stand out and arrows generally need to be larger (only the TMFS-6 arrows really stand out). In general, it would be better if labels were bold; those imposed on dark areas of thin sections should be white. In short, it is not possible to glean all of the important information about an image from the figure and caption alone. Because the data acquired from thin sections are important to this story, I suggest a different approach. Move the partial captions beneath each image into the main caption and add information. For example: TMFS-1 (20 porosity), TMFS-2 (15-20% porosity), and TMFS-3 (30-35% porosity) are all well sorted, subrounded, subarkoses with local poikilotopic calcite cement. Only TMFS-3 includes deformation bands. Porosity is reduced to <5% in the deformation band, within which small, angular grain fragments provide evidence of limited cataclasis (example highlighted with a bold arrow). Walk the reader through the rest of the photomicrographs in a similar way. Be sure to clearly state what you see as well as what you infer. You don't see pressure solution; you infer it from embayments in grains at point contacts (which can be better highlighted with bold arrows). You don't see cataclasis, you infer it from angular grain fragments. You don't see compaction; you infer it from reduced porosity and preferred alignment of elongate grains (which you don't mention anywhere, but should). In other photomicrographs you can see alignment of elongate clasts parallel to cross laminae or deformation bands. It's good to point that out. Also, I personally like the fact that you haven't drawn lines over deformation band boundaries. For readers less familiar with what these features look like, you may wish to provide some guidance in either words or arrows that mark top and bottom boundaries to a band.

Author Response – We have now separated this photo montage into two separate images (now Figs 9 and 10) showing Host Rock and Deformation Band examples respectively. Individual photos are now larger, and we have increased the font size and added a yellow fill to the labels so they are clearer. We've also added edge markers for the deformation bands in Fig. 10. Additional details are now in the Figure caption, rather than embedded in the figure. New figures attached.

Reviewer Comments – You note 'indistinct "fuzzy" boundaries to larger grains' beneath your last photomicrograph. Most of the grains are quartz and have sharp margins. Your labeled plagioclase grain has "fuzzy" margins, which are also locally brownish in color. Without being able to zoom in further or look at this on an SEM, I would say that there are several things that could contribute to this appearance. Top on my list is margins that are oblique, rather than perpendicular, to the surface of the thin section. Where the edge of a grain dips away from the grain center, it will be increasingly out of focus with distance. With extensive cataclasis, you may be looking through a zone of fine grain fragments on that grain edge. I think this is what you are referring to, but I'm not sure. If it is, spell it out and highlight the specific margin. If I were you, however, I would focus on more obvious evidence of cataclasis: a high percentage of angular grains that are substantially reduced in size with respect to subrounded grains evident in host rock.

Author Response – We agree there may be a number of reasons for seeing "fuzzy" edges to grains. However, the examples in question appear to be associated with feldspar grains, while adjacent quartz grains show very clear distinct edges. We make the observation, but have not expanded this

in any detail, and yes, have made more effort to emphasise the basic key observations, both in text and figure captions.

Reviewer Comments – On p. 8, you also discuss 'early development of sub-grain boundaries', and follow that on p. 9 with observations of 'clear sub-grain boundaries parallel to deformation band orientations.' In general, we use the term 'subgrain' (with no hyphen) to refer to a part of a larger grain separated from the host grain by a dislocation wall. Production of subgrains is part of the process of rotation recrystallization; it is not a brittle process. Subgrain boundaries are only visible with crossed polars, which causes differences in orientation of the crystal across these dislocation walls to show up as differences in exteniction (grayscale). The only features I see oriented subparallel to deformation bands appear to be cracks. Please revise the text for clarity and accuracy.

Author Response – We've added an XPL image to show an example of this (Fig 10d). There are only a few examples, and by far the dominant process is brittle (intragranular cracks and fractures, and shear fractures); however, we felt it was worth highlighting that this more plastic deformation was also apparent.

Reviewer Comments – Add a reference to the list of studies of deformation band impacts on flow (line 40): Sigda, J.M., Goodwin, L.B., Mozley, P.S., and Wilson, J.L., 1999, Permeability alteration in small-displacement faults in poorly lithified sediments: Rio Grande rift, central New Mexico: In Haneberg, W.C., Mozley, P.S., Moore, J.C., and Goodwin, L.B. (eds) Faults and Subsurface Fluid Flow in the Shallow Crust, AGU Monograph 113, 51-68.

Author Response – Done

Reviewer Comments – Change lines 51-53: "Deformation bands preferentially form in more poorly lithified layers within quartz arenite to arkosic sandstones (i.e. those lacking in lithics) at shallow depths (1–3 km; Fossen, 2010)" to: "Deformation bands within quartz arenite to arkosic sandstones (i.e. those lacking in lithics) preferentially form in more poorly lithified layers at shallow depths (1–3 km; Fossen, 2010)." The former suggests deformation bands are restricted to poorly lithified layers of specific composition.

Author Response – Done

Reviewer Comments – In lines 195-196, the authors refer to 'cycles enclosing blocks' and note common features of 'networks with lots of cycles'. The discussion of cycles refers to Figs. 6c and 2b&c, but it is not possible to understand how the reader is supposed to connect this information to the images. The term 'cycle' is not defined, and it is never mentioned again. If it is important to the story, the authors should define what they mean and why it is relevant. If it is not, they should remove references to 'cycles'.

Author Response – This was a term used in past publications describing the general methodology. We have now removed it here and replaced it with branches for consistency. i.e. branches bound an isolated segment.

Reviewer Comments – In line 218, the authors refer to 'a slightly coarser grained bed within the sandstone horizon'. I am not aware of a definition of 'horizon' used in this context. It appears to be a way to suggest associations between samples collected. Does it refer to the 4 m thick section of sandstone shown in Fig. 2? Please clarify.

Author Response – Replaced 'horizon' with 'unit', which we then introduce earlier to describe the sandstone unit sampled.

Reviewer Comments – I would like to see the authors replace references to 'weak' deformation or cataclasis with more specific information regarding observations rather than interpretations. I suspect they mean that evidence of fracture and associated grain-size and porosity reduction is present, but not as extensive as in other samples, as suggested by higher estimates of porosity.

Author Response – Done

Reviewer Comments – I suggest the authors replace 'grain crushing' with 'distributed microcracking' in places like line 233. I think it is a more accurate representation of the variable amounts of grain-size reduction via fracture illustrated in their thin sections. Their photos show a range from deformation bands in which the majority of grains are subrounded and similar in size to those in the host rock to deformation bands in which most of the grains have been reduced to relatively small angular fragments and relatively few original grains remain.

Author Response – Done, and have also added additional text within the figure captions for the microstructure figures (now Figs 9 & 10).

Reviewer Comments – On line 235, replace 'Calcite is also present' with 'Calcite locally fills pores'.

Author Response – Done

Reviewer Comments – Line 245 refers to early development of subgrain boundaries. I addressed misconceptions re: subgrain boundaries in the previous section on Specific Comments Linked to Figures. The authors should make appropriate changes to the text here also.

Author Response – We've added some XPL images to the microstructure figures which show that some higher strained quartz grains within deformation bands do appear to exhibit sub-grain boundaries (e.g. Fig 10d), though this is not a common feature.

Reviewer Comments – On line 248, the authors discuss embayed contacts. I think it would be helpful to clarify what is meant by 'embayed', with reference to more clearly annotated examples in thin section images.

Author Response – Added notes on Figs 9 and 10.

Reviewer Comments – The sentence beginning on line 256 states that 'Haematite is also incorporated into the matrix within deformation bands as a result of quartz grain crushing. Note the brownish-staining of deformation bands in Figs. 7a and 8". What evidence supports this interpretation? Is it possible that hematite was precipitated after formation of deformation bands? Please provide evidence (: : :and you don't need to hyphenate brownish & staining).

Author Response – Sentence removed as not relevant.

Reviewer Comments – Line 305 refers to 'minor cataclasis as evidence for shear'. Minor cataclasis can occur by compaction alone. It doesn't require shear.

Author Response – Re-worded.

Reviewer Comments – On line 309, the authors propose that evidence of compaction in sandstone suggests confining pressure may increase with proximity to the intrusion. It is certainly a sign of shortening, consistent with intrusion, but that suggests an increase in margin perpendicular stress, not an increase in confining pressure. Note also that intrusions, particularly shallow crustal intrusions, cool very rapidly. The temperature gradient between thin sheets of partially crystalline magma and wall rock so shallow it still has high porosity is very high, and temperature dissipates rapidly at cool shallow temperatures. If you know the thickness of individual sills and likely depth of intrusion, you can do a back of the envelope calculation to determine the cooling rate for a normal geothermal gradient (or even a slightly elevated one, which would not produce high temperatures at relatively shallow depths where you see high porosity sandstones). 'Pressure solution' actually has nothing to do with pressure. It is caused by a stress-induced chemical potential gradient. I suspect that what you are seeing is that the deformation bands that have accommodated the greatest deformation have the highest number of high-stress point contacts between grains, where solution mass transfer is facilitated.

Author Response – Re-worded in line with reviewer's comments.

Reviewer Comments – Line 311: Crush breccias do not consist of fragments that are only visible with a microscope. You do show clear evidence of cataclasis, which could be defined as distributed

microcracking and rotation and translation of resulting clasts. You might want to provide a definition like this where you first introduce the term in the paper, to facilitate discussion here.

Author Response – Re-worded in line with reviewer's comments, and also added additional wording in the figure captions in Figs 9 & 10 (microstructures).

Reviewer Comments – Lines 313-314: If a principal slip surface or fault core were present, you would call it a fault or a deformation band fault and not a deformation band.

Author Response – Re-worded to make the point that we are discussing deformation bands and not faults in this study, but that deformation band faults are observed in elsewhere on the intrusion.

Reviewer Comments – Line 317: I am not familiar with the term 'permeability pathway'. Deformation bands are features that can influence flow pathways, but they cannot be considered in isolation. In this case, the elephant in the hydrologic room is the extremely low permeability intrusion. Regional flow will take the 'easiest' route around the intrusion, which will be influenced not just by deformation band distribution and connectivity but also by the permeability of the surrounding undeformed rocks. This paragraph also reflects a misunderstanding of the hydrologic significance of microstructural observations. The fundamental misapprehension is that tabular structures that formed by different processes (e.g., compaction bands vs. cataclastic deformation bands) can influence flow differently even if they have the same permeability.

Your intrusion is effectively an impermeable wall. Your deformation band networks may redirect or inhibit flow in a shell around the plutons, or the main effect may be created by the intrusion itself. The best way to determine these effects would be to measure the permeabilities of cores cut in different directions through the networks, then work with a hydrogeologist to model flow. Without those data, I think you are restricted to providing a clear description of the structures at different scales. Please appreciate that the description itself is a significant contribution.

Author Response – We've modified the terminology here to state 'permeability and flow pathways', and made subtle changes to the wording in the paragraph. We've also added additional sentences to address the "elephant" that is the intrusion itself! Good point well-raised.

With regards to any misunderstanding of processes, we'd like to highlight that we have simply referred to points raised by other authors which have suggested that different deformation band types may impact fluid flow in subtly different ways. We agree that if two bands have the same permeability, then they will of course impact fluid flow in the same way. The point being made here is that two bands with the same porosity reduction may not have the same permeability due to different microstructures.

Reviewer Comments – Lines 327-328: I do not know if anyone has published evidence of magmatic fluids of appropriate composition to precipitate calcite. I think this would be a more compelling suggestion if the authors could cite a study indicating it was possible. I do know that the solubility of

calcite decreases with increasing temperature, so I suspect that heat introduced by the intrusion could facilitate precipitation of calcite from surrounding groundwater of appropriate composition.

Author Response – Re-worded to emphasize this latter point, which is what we were envisaging rather than the fluids being magmatically derived.

Reviewer Comments – Lines 335-336: I think it is particularly important to replace 'fractures' with 'deformation bands' in these sentences.

Author Response – Done

Reviewer Comments – Line 342: I don't know what the authors mean by "However, this assumes a homogeneous development of the grain-scale processes". Please explain.

Author Response – We've added additional wording here to highlight that the application of the Kozeny-Carmen equation here is an over-simplification as deformation bands are intrinsically heterogeneous!

Reviewer Comments – In line 351, the authors state "At Trachyte Mesa, deformation bands decrease markedly from ~5 to 10m above the intrusion margin.." I assume they mean deformation band intensity decreases. However, they do not present data showing vertical variations in deformation band networks. Is this a personal observation? If the authors have data that show this variation, they should provide it. If relevant data have been published, they should cite the reference.

Author Response – Yes, this is a personal observation, but as stated, outcrops are limited, so detailed analysis may be challenging. The purpose of making this observation was to bridge the discussion. Adding additional data/ figures may detract from the key messages in the paper, particularly as the vertical variations have not been analyzed to the same extent as the horizontal variability. We agree this is an interesting area for further analysis, but we do not currently have the data available to expand on this further right now.

Reviewer Comments – I suggest the authors modify lines 354-357 to state: "In addition to reducing the bulk permeability of the reservoir, the deformation bands largely strike parallel to the intrusion margin (Wilson et al., 2016), producing an anisotropy in permeability similar to that of a fault zone (e.g. Farrell et al., 2014).

Author Response – Done

Reviewer Comments – Line 360-361 should be modified to state: 'Gaining a better understanding of these emplacement-related deformation structures may have important implications for fluid flow, hydrocarbon reservoir connectivity / deliverability, hydrology, geothermal energy and CO2 sequestration: : :'

Author Response – Done

Reviewer Comments – I suggest the authors modify line 378-379 as follows: "The increase in margin-parallel Y- and X-nodes with proximity to the intrusion is likely to inhibit flow perpendicular to the intrusion margin, as well as potentially forming non-producible reservoir zones."

Author Response – Done